# Structural basis for the interaction between the *Drosophila* RTK Sevenless (dROS1) and the GPCR BOSS

Jianan Zhang[1,2,3], Yuko Tsutsui [1,2,3], Hengyi Li[1,2,3], Tongqing Li [1,2], Yueyue Wang[2], Salma Laraki[1,2], Sofia Alarcon-Frias[1,2], Steven E. Stayrook [1,2] & Daryl E. Klein [1,2]✉

Sevenless, the *Drosophila* homologue of ROS1 (University of Rochester Sarcoma) (herein, dROS1) is a receptor tyrosine kinase (RTK) essential for the differentiation of *Drosophila* R7 photoreceptor cells. Activation of dROS1 is mediated by binding to the extracellular region (ECR) of the GPCR (G protein coupled receptor) BOSS (Bride Of Sevenless) on adjacent cells. Activation of dROS1 by BOSS leads to subsequent downstream signaling pathways including SOS (Son of Sevenless). However, the physical basis for how dROS1 interacts with BOSS has long remained unknown. Here we provide a cryo-EM structure of dROS1's extracellular region, which mediates ligand binding. We show that the extracellular region of dROS1 adopts a folded-over conformation stabilized by an N-terminal domain comprised of two disulfide stapled helical hairpins. We further narrowed down the interacting binding epitopes on both dROS1 and BOSS using hydrogen-deuterium exchange mass spectrometry (HDX-MS). This includes beta-strands in dROS1's third Fibronectin type III (FNIII) domain and a C-terminal peptide in BOSS' ECR. Our mutagenesis studies, coupled with AlphaFold complex predictions, support a binding interaction mediated by a hydrophobic interaction and beta-strand augmentation between these regions. Our findings provide a fundamental understanding of the regulatory function of dROS1 and further provide mechanistic insight into the human ortholog and oncogene ROS1.

*Drosophila* ROS1 (dROS1), also known as Sevenless, is a critical RTK required for R7 photoreceptor development[1]. Sevenless has been a cornerstone in the study of *Drosophila* eye development and has significantly contributed to our understanding of cell differentiation and signaling pathways. Genetic screens involving Sevenless have led to the identification of key components of the Ras/MAPK signaling cascade, including Son of Sevenless (SOS), which is a guanine nucleotide exchange factor activating Ras1, and downstream kinases[2]. Interestingly, rather than a soluble morphogen or growth factor, the ligand for dROS1 is a membrane-bound GPCR, which is

an unusual ligand for an RTK[3]. The first reported physical interaction between dROS1 and BOSS was a heterotypic cell-aggregation assay involving dROS1-expressing and BOSS-expressing S2 cells, and further supported by co-immunoprecipitation[3,4]. It has been proposed that the positional proximity of dROS1 on noncommitted R7 precursor cells with BOSS on adjacent differentiated R8 cells promotes binding through cell-cell contacts[5]. However, no quantitative in vitro study has been performed to confirm nor has any structural study been done to shed light on the details of this unusual interaction.

[1]Department of Pharmacology, Yale University School of Medicine, New Haven, CT 06520, USA. [2]Yale Cancer Biology Institute, Yale University, West Haven, CT 06516, USA. [3]These authors contributed equally: Jianan Zhang, Yuko Tsutsui, Hengyi Li. ✉e-mail: daryl.klein@yale.edu

**Fig. 1 | The N-terminal region of dROS1 interacts with BOSS' ECR. a** Illustration of the architecture of dROS1 and BOSS. The large extracellular region of dROS1 is composed of 9 Fibronectin type III (FNIII) domains (number 1–9, brown rectangles) and 3 YWTD propellers (A-C, green hexagons). BOSS is a GPCR with an extracellular region and 7 transmembrane helices. The name designation for constructs used in this study are based upon terminal FNIII domains. **b** Representative Coomassie stained SDS-page gel analysis of different dROS1 ECR constructs. **c** Biacore T200 binding analysis of dROS1's ECR constructs to BOSS' ECR. The minimal construct of dROS1 (2-3, second to third FNIII domain) binds to BOSS as well as the full ECR (1–9) (*n* = 3 biologic repeats for all three constructs; error bars on all points represent standard deviation across all independent experiments). One-way ANOVA test shows no significant difference among means (*P* > 0.05). $K_D$ values are shown as mean ± standard deviation (error bar) from individual experiments.

Like all other RTKs, the architecture of dROS1 is composed of an extracellular region (ECR), a single-pass transmembrane domain (TM), and an intracellular kinase domain (KD). Notably, ROS1 has the largest ECR of any described RTK. Human ROS1 and dROS1 both are predicted to have an ECR composed of nine fibronectin type III domains (FNIII) and three YWTD beta-propellers (Fig. 1a). Importantly, however, no structural studies have been done on ROS1 from any species to date. Therefore, there is no experimental evidence for the ECR architecture of ROS1. Interestingly, dROS1 contains a poly-arginine (9 consecutive arginines) sequence in the last FNIII domain just prior to the transmembrane domain. This sequence harbors multiple consensus furin proteolytic cleavage sites (i.e., RXRR) (Fig. 1a). Notably, cleavage occurs topologically in the same FNIII loop where insulin receptor is processed[6]. Previous literature suggests that the large ECR of dROS1 is cleaved at this position but remains non-covalently associated with the rest of the receptor[7]. This is in contrast to the insulin receptor which is cleaved but remains covalently associated through disulfide bonds to the kinase domain[8,9].

Surprisingly, the soluble ECR of BOSS is insufficient to activate– and paradoxically inhibits–dROS1 even when presented as a pre-formed oligomer[4]. This is contrary to the 'central dogma' of RTK oligomer-induced activation[10,11]. However, a dROS1 mutant that removes most of the ECR is constitutively active[12]. While over-expression of many RTKs can lead to ligand-independent activation due to increased receptor dimerization, overexpression of dROS1 or a chimeric dROS1 with the kinase domain of dEGFR (Drosophila Epidermal Growth Factor Receptor) does not result in activation in the absence of ligand, revealing a potent inhibitory role of dROS1's ECR[1]. Together, this points to a unique regulatory mechanism employed by dROS1 unlike that of other well-studied RTKs. To further investigate this activation mechanism, we solved the structure of dROS1's ECR and analyzed the biophysical basis for its interaction with BOSS.

Here we show, the extracellular region of dROS1 adopts a folded-over conformation stabilized by an N-terminal domain comprised of two disulfide stapled helical hairpins. Using hydrogen-deuterium exchange mass spectrometry (HDX-MS) we identified the interacting binding epitopes on both dROS1 and BOSS. An AlphaFold prediction of their complex, supported by HDX and mutagenesis studies, details the interaction between dROS1 and BOSS necessary for regulating dROS1's

activation. Together, our findings provide a model that sheds light on the regulatory mechanisms governing dROS1 activation.

## Results

### The N-terminal region of dROS1 interacts with the GPCR BOSS

To characterize the binding interaction between dROS1 and BOSS, we purified the full-length ECR of dROS1, including FNIII domains 1 through 9, construct (1-9) Fig. 1a, b—as well as smaller ECR fragments (Fig. 1b)—and examined their binding affinities towards the ECR of BOSS using surface plasmon resonance (SPR, Biacore) (Fig. 1c). We found that the full ECR of dROS1 binds to BOSS with a $K_D$ of 4.4 μM (Fig. 1c). RTKs with soluble ligands typically exhibit nanomolar binding affinities or require co-receptors to capture dilute signals[11]. In contrast, dROS1 binds to BOSS with a comparatively modest micromolar affinity. Since BOSS is also a membrane-bound receptor, this binding affinity is consistent with that of other cell-cell adhesion receptors (i.e. notch and delta[13]) where both ligand and receptor are spatially restricted. We then determined that the N-terminal region of dROS1 (constructs 1-4 and 2-3) also binds to BOSS with a similar affinity (Fig. 1c). Thus, these truncations narrow down the binding epitope on dROS1 to the region flanked by the second and third FNIII domains (construct 2-3) and includes the first YWTD propeller domain.

### The N-terminal region of dROS1 adopts a folded-over conformation

To establish a biophysical understanding of dROS1 function we sought to obtain a 3-Dimensional structure of dROS1's ECR. Notably, there is no published structural information on the ECR of ROS1 from any species. We expressed and purified the dROS1 ECR (construct 1-6) (Fig. 2a) and obtained a 3.94 Å structure using single-particle CryoEM (Fig. 2b; Supplementary Fig. 1). Surprisingly, model building only resolves the N-terminal region (equivalent to construct 1-4) starting from the N-terminus to the fourth FNIII domain (Fig. 2a, b). The C-terminal domains starting from the fifth FNIII domain to the sixth, including the intervening YWTD-C, were not observed in the map (Fig. 2a, b). The lack of density for the C-terminal domains is consistent with this region demonstrating greater flexible with respect to the N-terminal domains. The poor density and low resolution of the fourth FNIII domain also support this idea (Supplementary Fig. 1). Despite the

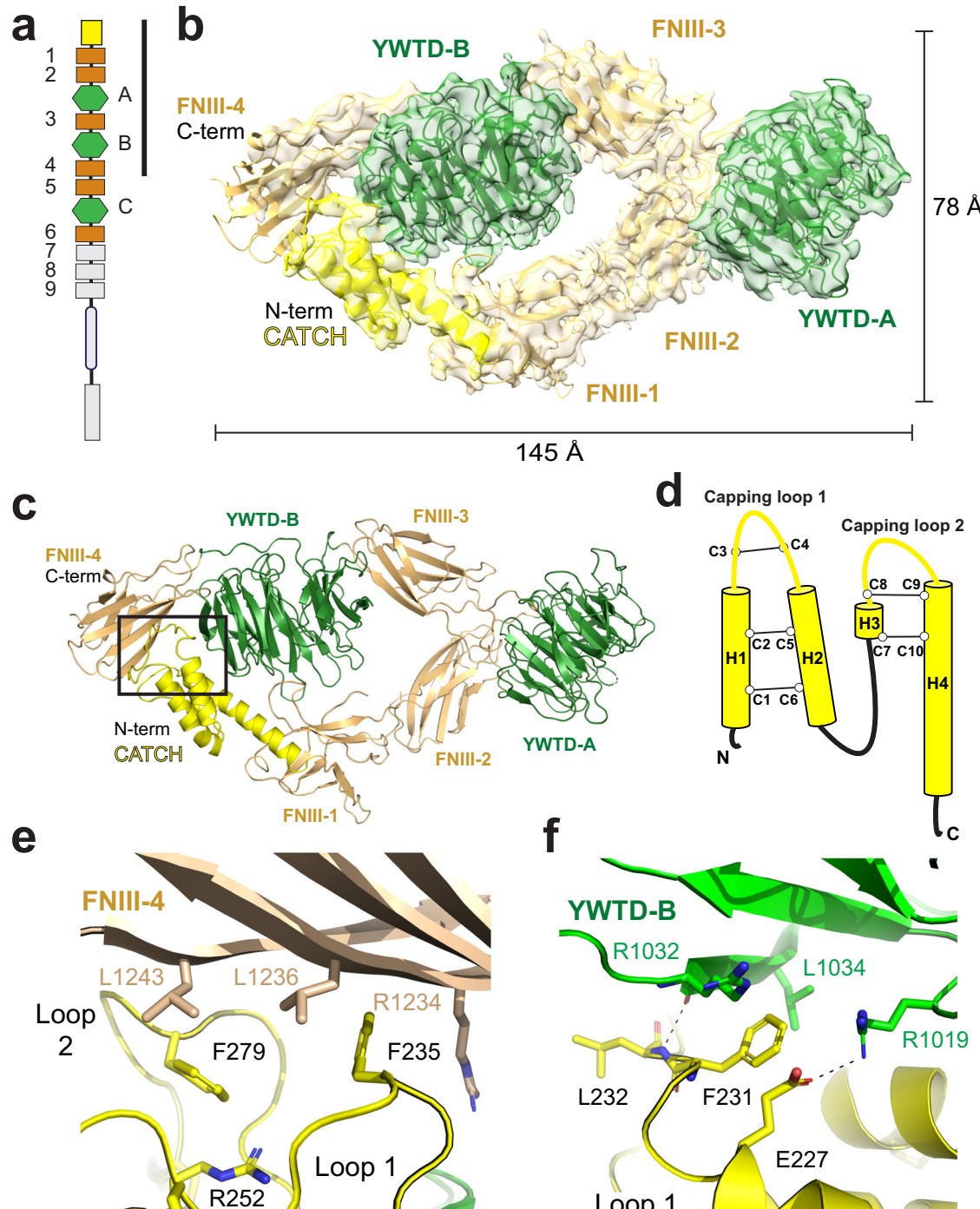

**Fig. 2 | CryoEM Structure of the N-terminal ligand binding region of dROS1.**
**a** Schematic of the construct used for CryoEM. The colored domains were expressed and purified. The reconstruction only resolves up to the fourth FNIII domain, designated by a vertical black line. **b** Cartoon representation of dROS1 ECR structure and fitted into the CryoEM map at 3.94 Å resolution, with FNIII domains in light brown and YWTD domains in green. Only the N-terminal domains (equivalent to construct 1–4) of dROS1's ECR can be resolved in the reconstructed map. The N-terminal ECR of dROS1 adopts a folded-over conformation held by the interaction of the CATCH domain (colored in yellow) with the second propeller (YWTD-B) and fourth FNIII domain (FNIII-4). **c** Cartoon representation of dROS1's ECR structure. **d** Topology and disulfide linkage of the CATCH domain. **e** Zoomed-in view of the interaction of the N-terminal CATCH domain with the fourth FNIII domain, as boxed in panel (**c**). F235 in capping loop 1 and F279 of loop 2 direct a hydrophobic interaction with FNIII-4. **f** The N-terminal CATCH domain interacts with YWTD-B shown as close-up of region boxed in panel (**c**). The interactions center around F231 of capping loop1 of the CATCH.

flexibility, we were able to further improve the density of the fourth FNIII domain by masked local refinement (Supplementary Fig. 2).

Interestingly, the model reveals that rather than having a linear arrangement of domains, the N-terminal region adopts a folded-over conformation stabilized by an interaction of an N-terminal cysteine-rich region with the second propeller (YWTD-B) and fourth FNIII domain (Fig. 2b, c). The N-terminal cysteine-rich region forms a previously unappreciated domain (Fig. 2a–c, yellow). The domain is comprised of two disulfide stapled helical hairpins that associate with one another through hydrophobic interactions (Fig. 2c, d). The two

helical hairpins generate two loops (capping loops 1 and 2) which stabilize the N-terminal ECR compacted fold. Despite the rather limited resolution in this region, we observe clear density for bulky side-chains which assisted in model building (Supplementary Fig. 3).

In the loop generated by the first helical hairpin, capping loop 1, F235 sits between R1234 and L1236 of the fourth FNIII domain (Fig. 2e). The loop generated by the second helical hairpin, capping loop 2, also interacts with the fourth FNIII domain with residue F279 sitting between L1243 (FNIII-4) and R252 of the first hairpin (Fig. 2e). Using a similar interaction, loop 1 also makes contact to the YWTD-B domain. F231 sits between R1032 and L1034 of the second propeller. This interaction is further supported by a main chain hydrogen bond between L232 and R1032 (Fig. 2f). Notably, these surface exposed hydrophobic residues and basic residues are well-conserved across different insect species, suggesting the importance of the capping interactions and conservation of the folded-over conformation (Supplementary Fig. 4).

The disulfide stapled helical hairpin structure of the N-terminal cysteine rich region of dROS1 is unusual. However, it reminded us of a structure that we recently solved of the related RTK ALK (Anaplastic Lymphoma Kinase) bound to its ligand ALKAL (ALK And LTK Ligand)[14]. Indeed, the unique cysteine-rich region of dROS1 structurally resembles ALKAL (Supplementary Fig. 5). Both share the rare fold of a doubly disulfide-stapled helical hairpin. When the helical hairpins of dROS1 and ALKAL are structurally aligned, it is evident that they utilize similar surfaces to engage with their respective receptors (Supplementary Fig. 5b, d). This alignment highlights a convergent structural and functional strategy. Therefore, from this perspective, dROS1 could be envisioned as a self-ligated receptor through the N-terminal cysteine rich region, stabilizing a bent-over conformation. Given the relatedness of this region to ALKAL and its role in stabilizing the dROS1 N-terminal ECR, we designate it as a CATCH domain for (Cysteine-rich ALKAL-Type Coupled Helices).

## Structural prediction of BOSS reveals a "Venus-flytrap" architecture similar to mGluRs

The evolution of the orphan GPCR BOSS diverged so rapidly that tracing homologs by sequence is incredibly difficult[15]. However, the sequence of BOSS' transmembrane segments suggests homology to metabotropic glutamate receptors (mGluRs) among Class-C GPCRs[15]. An AlphaFold[16–19] prediction of BOSS' ECR suggests it adopts a bi-lobed "Venus-Flytrap" domain (VFTD) architecture, also similar to mGluRs (Fig. 3a–c). A Dali[20,21] structural comparison search of the predicted structure shows the highest similarity to mGluR-2 (Supplementary Fig. 6).

When compared with mGluR-2, BOSS' ECR reveals several unique features. BOSS lacks a Cysteine-rich domain (CRD) "stem" region found in mGluR-2 that separates the VFTD and membrane region (Supplementary Fig. 7a). Instead, BOSS is predicted to have a configuration parallel to the membrane where after Lobe 2 a peptide leads to the transmembrane region (Fig. 3b, c). This includes a linker peptide (residues 480–500) following strand 7 of Lobe 2. A cysteine at 501 forms a disulfide bond restraining this peptide to Lobe 2. However, this peptide has a very low pLDDT (per-atom confidence estimate) score indicating a poor model−consistent with a dynamic and flexible peptide (Fig. 3b, c).

## BOSS lacks glutamate binding and receptor dimerization motifs

Notably, while BOSS ECR shows structural similarity to the VFTD of mGluRs, it does not retain many of the functionally important residues. When the first Lobe of BOSS and mGluR-2 are aligned, it is apparent that the key residues for binding glutamate are not retained (Fig. 3d). Indeed, Helix-A of Lobe 1 does not extend far enough to reach the glutamate binding pocket in BOSS – lacking the critical arginine (R61) for glutamate binding (Fig. 3d, orange portion Helix-A). Additionally,

BOSS completely lacks a structural loop feature (residues 166-186) in mGluRs that forms the glutamate binding pocket (including A166 and T168) (Fig. 3d, orange). Rather, in BOSS the first helix of Lobe 2 arises directly from the beta strand in Lobe 1 that precedes the glutamate binding structural loop. Importantly, the first helix of Lobe 2 positions the entire Lobe 2 (as a rigid body) with respect to Lobe 1. This helix moves after glutamate binding in mGluRs – moving from an "open" to "closed" state (Fig. 3d, black to gray). BOSS' first helix of Lobe 2 is found in neither of these positions, and is >15 Å away, passing directly through the glutamate pocket (Fig. 3d). Therefore, BOSS has a unique orientation of Lobe 2 and could be considered as "pocketless".

Furthermore, the residues that dictate dimerization of mGluRs are also not conserved in BOSS. In mGluR-2, dimerization is directed by Helix-B and Helix-C of Lobe 1 (Supplementary Fig. 7a, b). When structurally aligned, the hydrophobic nature of the mGluR-2 dimer interface is not maintained in BOSS (Supplementary Fig. 7b). Indeed, Helix-C is smaller in BOSS and lacks the terminal L157 and F158 equivalent residues that are crucial for dimerization (Supplementary Fig. 7c, orange). Together, this indicates that despite sharing a Venus-flytrap fold, BOSS' function is distinct from that of mGluRs.

## Mapping the binding interfaces in the dROS1-BOSS complex using HDX-MS

To find the binding interface for the interaction between dROS1 and BOSS, we used hydrogen-deuterium exchange and mass spectrometry (HDX-MS). HDX-MS probes protein conformation and protein-protein interactions by assessing deuterium uptake of the amide hydrogen atoms of the protein backbone[22]. By comparing deuterium uptake of peptide regions between each monomeric protein and the complex ($\Delta$Ds(i))[23], potential binding regions can be identified (Fig. 4). We have previously used this technique to map the binding epitope of antibodies targeting the RTK ALK[14]. For this experiment we used the dROS1 (1-4) construct and the full-length BOSS ECR. These constructs were stable at the concentrations necessary for the experiment and include the interaction sites (Fig. 1c). Given that the affinity between these proteins is relatively weak (~3–5 μM $K_D$), we developed a strategy to preserve the interaction by immobilizing His-tagged protein on Ni-NTA resin to increase the local concentration (details in Methods). To aid in visualization, regions protected from exchange (i.e. stabilized regions that are candidate binding epitopes) were mapped onto the structures of dROS1 and BOSS (Fig. 4b, d, Supplemental Fig. 8, Supplemental Table 2).

Regions protected from deuterium exchange in dROS1 upon BOSS binding are found throughout dROS1 (Fig. 4a, b, blue). However, within the determined binding region of dROS1 (second through third FNIII domains) (Fig. 1c), significantly protected regions are localized in the third FNIII domain. This includes the first strand of the domain (residues 814–838), as well as the third and fourth strands (residues 876–896). Between these peptides, peptide 878-896 is more surface exposed and thus more likely to be a direct interaction site (Fig. 4a, b, blue).

HDX-MS results show that one of the significant regions in BOSS upon dROS1 binding includes the C-terminal beta-strand "FVTNVTTY" (residue 471–478, Fig. 4c, d, blue). This corresponds to strand 7 of Lobe 2 (Fig. 3c). Another stabilized region in BOSS includes the central linker that spans Lobe 1 and Lobe 2−beginning with Helix-B of Lobe 1 and passing through to the central first Helix of Lobe 2 (residues 220–239) (Fig. 4c, d, blue). This corresponds to the lobe linking peptide highlighted in Fig. 3d. As this peptide is mostly confined between lobes, it is unlikely to directly participate in dROS1 binding. Thus, based on our HDX results, aided with our structural prediction (Fig. 3), peptide 471–478 (C-terminal beta-strand "FVTNVTTY") may represent a direct binding epitope for dROS1. Together, our HDX-MS data combined with our structure and binding data narrows down the likely complex binding epitope to dROS1's third FNIII domain and a strand near BOSS' ECR C-terminal loop.

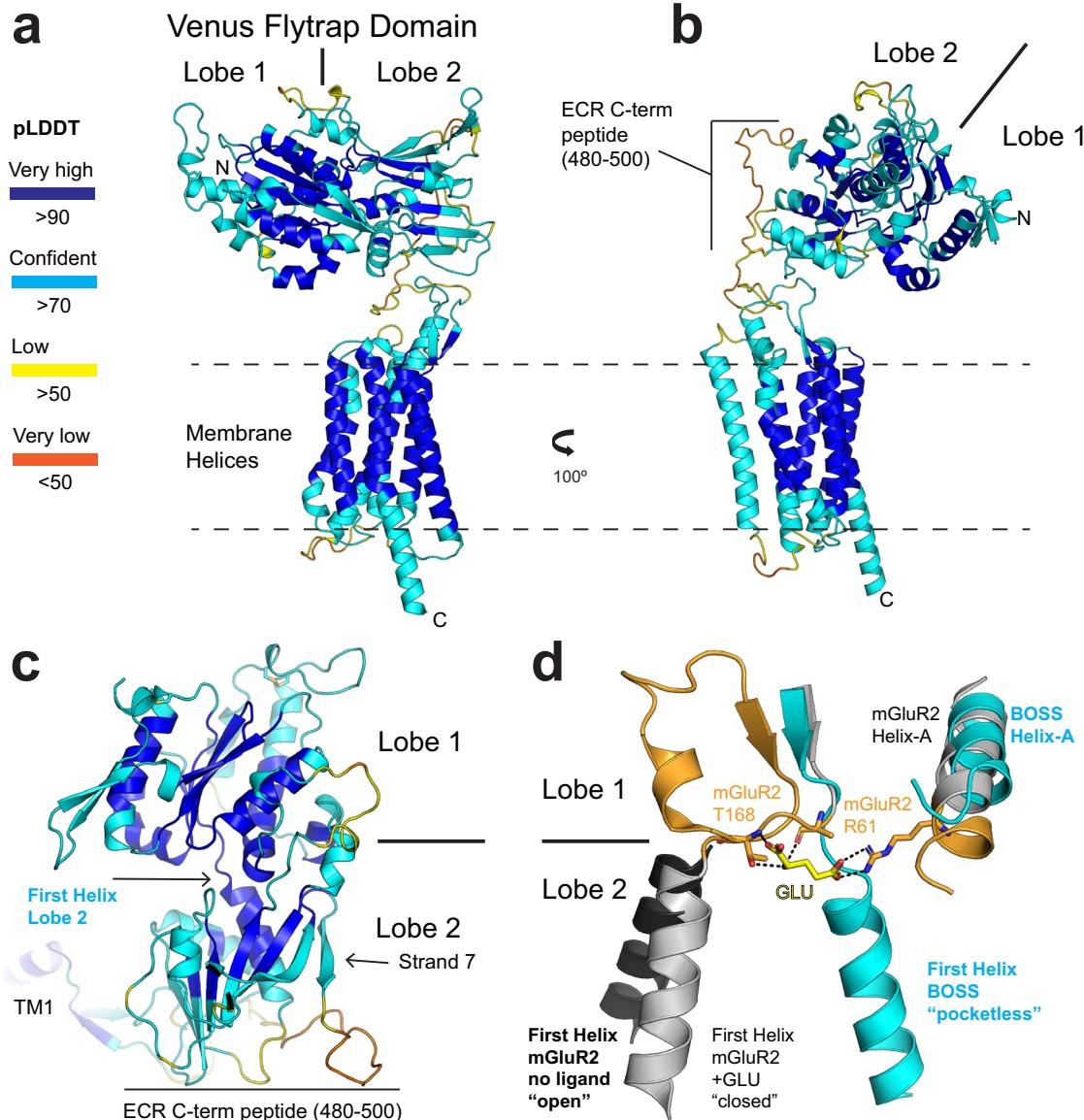

**Fig. 3 | BOSS' ECD structure is predicted to have a Venus-flytrap fold similar to mGluRs. a** Top-ranked model of BOSS' ECR generated from AlphaFold 3 colored by pLDDT (confidence in local structure) score. The ECR adopts a Venus-Flytrap domain (VFTD) with two lobes. The model predicts an overall orientation of the ECR to be parallel to the membrane. **b** A 100 degree rotation of view in panel (**a**). Notably, the ECR C-terminal peptide (480–500) has a low pLDDT score. **c** The ECR of BOSS in an upright orientation as commonly viewed for mGluRs. The central first helix of Lobe 2 is highlighted, as is the final ECR strand "strand 7" of Lobe 2. **d** An overlay of BOSS' ECR with mGluR-2, aligned to Lobe 1. BOSS is in approximately the same orientation as panel (**c**). Regions colored orange are not present in BOSS. This view emphasizes the glutamate (GLU) binding pocket of mGluR-2. Ligand bound mGluR-2 is shown in gray and orange (PDB ID: 5CNI)[48], with bonds to GLU shown. BOSS lacks the regions (orange) that are responsible for forming the binding pocket. The inactive (non-ligand bound) state of mGluR2 (PDB ID: 7epa)[49] is colored black. The orientation of BOSS' Lobe 2 with respect to Lobe 1 is not similar to either the Apo (black) or ligand bound (gray) state of mGluR-2. Instead, BOSS' first helix of Lobe 2 passes directly through the ligand binding pocket, rendering it "pocketless".

## Structural prediction of the dROS1-BOSS complex

In the absence of an experimentally determined dROS1-BOSS complex structure, we sought to generate a theoretical model of the complex using AlphaFold[16–19] (see Methods). Since our binding studies narrowed down the binding epitope to the N-terminal region of dROS1, we were able to apply this constraint when generating models. Initial predicted complex structures using AlphaFold 2 all narrowed in on third FNIII domain of dROS1. We then used AlphaFold 3[19]—which allows for post translational glycosylation—to generate a model of the target region. All AlphaFold 3 complexes converged with high confidence (ipTM=0.77, pTM=0.85) to a single model (Fig. 5a, b). Importantly, the predicted structure is consistent with our HDX-MS and binding experiments—the predicted complex interface includes the very

epitopes identified in our HDX-MS (Fig. 5c). In the model, the interaction of the complex is mediated by parallel beta-strand augmentation between dROS1's third FNIII domain (874-890) and a C-terminal strand of BOSS' ECR (474-491) (Fig. 5d).

In the complex, strand augmentation builds on dROS1's third FNIII's strands (873-890) by coupling to two strands on BOSS (474-491), in parallel. Notably, this region of BOSS (ECR C-terminal peptide) had a low pLDDT score (<50, orange) in the unbound state—consistent with disorder (Fig. 3b, c). In the model complexed with dROS1, this peptide generates a new strand with high pLDDT scores (>90, dark blue), indicating high confidence in the structure (Fig. 5b). This is consistent with the C-term peptide of BOSS going from a disordered to ordered structure as it mediates binding to dROS1. The packing of the

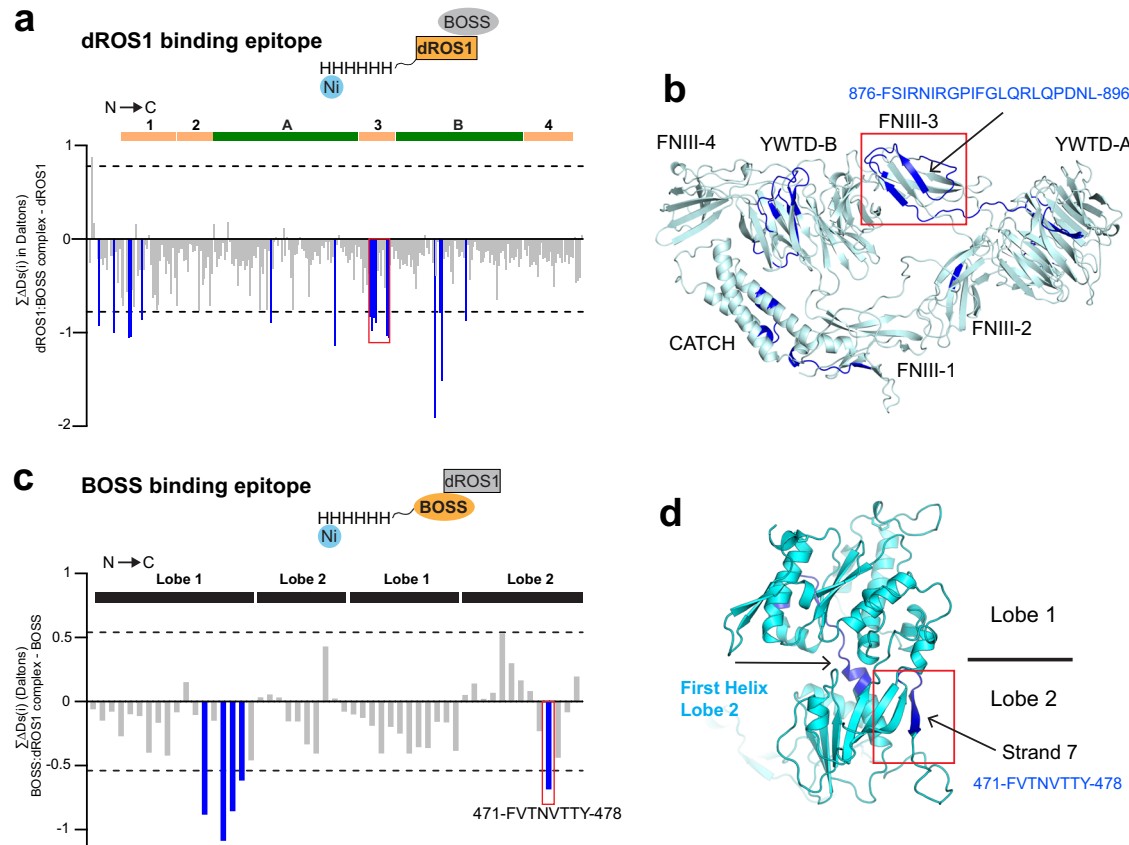

**Fig. 4 | The dROS1-BOSS interaction probed by HDX-MS. a** A butterfly plot of BOSS binding to immobilized dROS1 on Ni-resin. Each bar shows the sum of a deuterium uptake difference for a peptide between BOSS-bound dROS1 and dROS1 at 10 min deuterium labeling time (ΔDs(i), where i = assigned peptide number) from two biological repeat experiments (n = 2) as described and calculated previously[23]. The black dotted lines (Ds = ±0.7746 Da) show the 90% confidence limit for ΔDs(i) data. Blue bars show regions in dROS1 undergoing statistically significant structural changes upon BOSS binding. The bars at the top of the plot indicate the position of FNIII (light orange) and YWTD (green) domains. The peptide regions corresponding to the third FNIII-3 domain are indicated by the red box. **b** Possible binding interfaces indicated by HDX-MS in panel (**a**) are colored dark blue on the dROS1

structure. For the list of all analyzed peptide amino acid sequences, their raw deuterium uptake, and calculated ΔDs(i) values, see the *Source Data*. **c** A butterfly plot of dROS1 binding to immobilized BOSS on Ni-resin. Peptide regions with negative ΔDs(i) values show shielded regions from the deuterium solvent upon dROS1 binding. The black dotted lines (Ds = ±0.5409 Da) represent the 90% confidence limit for ΔDs(i) data with n = 2 biological repeats (3 technical repeats/n). The black bars at the top of the plot denote the VFTD lobe. Peptide regions with ΔDs(i) exceeding the confidence limit undergo statistically significant structural changes upon dROS1 binding and are shown as blue bars. **d** Possible binding interfaces indicated by HDX-MS in panel **c** are colored dark blue in an AlphaFold predicted BOSS structure.

beta-stand augmentation is supported by an extension of dROS1's FNIII's hydrophobic core. This is coordinated by BOSS' Leucine residues L489 and L490 making hydrophobic interactions with dROS1's FNIII core (F876, I878, F886 and L888) (Fig. 5d). In addition to main chain beta-strand interactions, there are multiple predicted side chain interactions that involve basic residues from dROS1 interacting with acidic residues on BOSS. This includes: R879 on dROS1 interacting with E465 of BOSS; R882 bonding with D484 on BOSS, and R890 bonding with E490 on BOSS (Fig. 5d).

We wished to test the predicted complex model using site directed mutagenesis. However, site specific mutagenesis to disrupt the complex interaction is difficult to generate as the interaction is mediated mainly by main chain strand augmentation. Mutation or deletion that overtly disrupts the strand's secondary structure would likely lead to misfolding of the domain itself. Therefore, we sought to disrupt the hydrophobic interactions mediated by dROS1's third FNIII domain including the peptide (F876-S877-I878), "FSI", with the Leucine residues on BOSS (L489-E490-L491), "LEL". Notably, the "FSI" residues are not conserved between the second and third FNIII domains in dROS1, which could explain the specificity for the interaction with the third FNIII domain (Supplementary Fig. 9a). Rather than the hydrophobic residues "FSI", the second FNIII domain has a hydrophilic

sequence "SEQ". Therefore, to disrupt the complex interaction by mutagenesis without disrupting the fold, we mutated the "FSI" of third FNIII domain into the corresponding residues "SEQ" found in the non-binding second FNIII domain. Binding results of dROS1$_{SEQ}$ demonstrate no observable binding to BOSS (Fig. 5e). This result is consistent with the model. However, the dROS1$_{SEQ}$ construct was less stable and less pure than the WT construct.

Therefore, to further validate the model, we sought to mutate the residues on BOSS that interact with the hydrophobic core of dROS1's third FNIII domain – the reciprocal to the dROS1$_{SEQ}$ experiment. Mutation of Leucine residues in BOSS (L489 and L491) to non-hydrophobic residues ("LEL" to "DEQ"; BOSS$_{DEQ}$) expressed as well as the WT indicating that the mutations did not affect protein stability. SPR analysis of BOSS$_{DEQ}$ binding to the full ECR of dROS1 (Construct (1-9)) shows no detectable binding (Fig. 5e). This finding is consistent with the model of the interaction between dROS1 and BOSS (Fig. 5a and c).

**Impact of binding site mutations on dROS1 cell activation**
To assess the impact of binding site mutations on the signaling of full-length dROS1 in response to membrane-embedded full-length BOSS, we created a chimeric dROS1 receptor with a human ALK kinase domain. ALK was selected because it is the RTK with the highest kinase

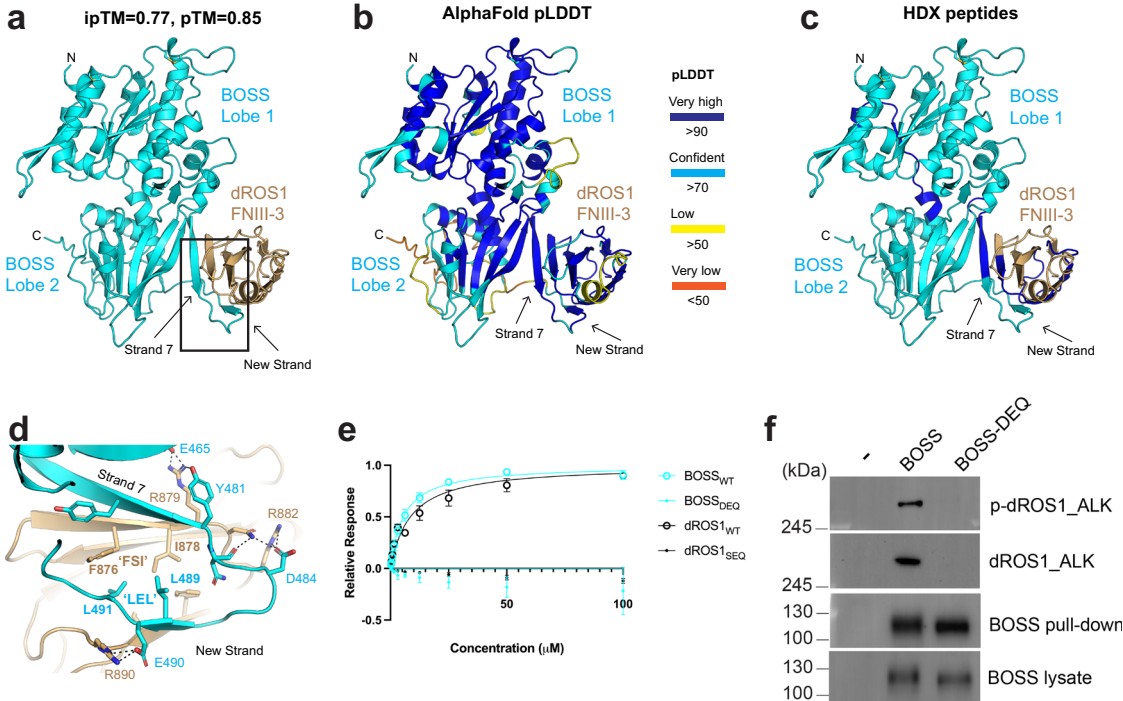

**Fig. 5 | Structural model of the complex between dROS1 and BOSS. a** AlphaFold 3 prediction of dROS1's third FNIII domain in complex with BOSS' ECR. The ipTM (confidence in protein-protein interaction) score above 0.7 indicates high confidence in the complex. The global protein (pTM) score is similarly high. The predicted model shows the third FNIII domain (FNIII-3) of dROS1 interacting with the C-terminal peptide of BOSS' ECR. **b** The model as in panel **a**, colored by the local structure confidence pLDDT score as shown. **c** The peptides identified by HDX-MS are colored dark blue in the complex model. **d** A close up view of the boxed region in panel (**a**). Strand 7 of BOSS' Lobe 2 mediates parallel beta-strand augmentation with dROS1. Interacting side chain residues are labeled. BOSS' hydrophobic Leucine residues (L489 and L491) of "LEL" interact with the hydrophobic core of dROS1's third FNIII domain including (F876 and I878) of "FSI". **e** Binding analysis shows

dROS1$_{SEQ}$ (mutation of F876 and I878) does not interact with BOSS ($n = 2$ biologic repeats for both dROS1$_{WT}$ and dROS1$_{SEQ}$). Similarly, mutation of BOSS' C-terminal peptide from "LEL" to 'DEQ' disrupts binding to dROS1 (construct 1-9) ($n = 3$ biologic repeats for BOSS$_{DEQ}$, $n = 2$ biologic repeats for BOSS$_{WT}$). Data are shown as mean +/− standard deviation (error bar) from individual experiments. **f** Cell activation assay of full-length BOSS and dROS1. A chimeric dROS1 construct with the kinase domain of ALK was used to track total and phosphorylated receptor. His-tagged full-length BOSS was transiently expressed in dROS1$_{ALK}$ stable cell. BOSS was pulled down with Ni resin, and probed with ALK kinase antibodies ($n = 3$ biologic repeats). BOSS-DEQ is the mutant of BOSS with L489 and L491 of "LEL" mutated to "DEQ". Representative blots are shown.

domain homology to ROS1, and both cluster within the same family[24]. This chimera enables us to track the levels of total and phosphorylated dROS1 receptor using well established antibodies targeting the ALK kinase domain, as in our previous studies on human ALK activation[14].

A stable NIH-3T3 cell line expressing the chimeric dROS1$_{ALK}$ receptor was established. To evaluate the ability of wild-type BOSS and the binding site mutant BOSS$_{DEQ}$ to stimulate dROS1$_{ALK}$ phosphorylation, we transiently expressed either BOSS or its mutant counterpart in the stable dROS1$_{ALK}$ cells. Cell lysates confirmed the expression of His-tagged BOSS in transiently transfected cells (Fig. 5f). BOSS was successfully pulled down using Ni resin, demonstrating enrichment of BOSS. The Ni-purified BOSS was probed for total and phosphorylated dROS1$_{ALK}$.

Wild-type BOSS was able to pull down dROS1$_{ALK}$, whereas the BOSS$_{DEQ}$ mutant did not (Fig. 5f). Additionally, phosphorylated dROS1$_{ALK}$ was detected only in the presence of wild-type BOSS but not with the mutant, consistent with the binding impairment caused by the DEQ mutation (Fig. 5f). Together, the binding and cellular studies support the structural model of the interaction between dROS1 and BOSS.

## Discussion

dROS1 is homologous to the human RTK and oncogene ROS1, which is the least well-studied RTK[10]. Multiple ROS1 fusion proteins have been discovered to play an important role in a variety of human cancers including glioblastoma, non-small-cell lung cancer, and breast

cancer[25,26]. Due in part to the difficulty of purifying ROS1, no structural insight has yet been established to reveal the architecture of its ECR region. In addition, ROS1 is the last orphan human RTK, with its ligand remaining to be firmly established. Also, a mammalian BOSS homolog has not been identified[5]. Thus, ROS1 is similar to the RTK ALK, where the ligand controlling its activation is not structurally conserved between invertebrates and vertebrates[27–30].

We find here that dROS1 is regulated by relatively weak interactions with BOSS that are enhanced by the high local concentration when restricted to two dimensions between juxtaposed cell membranes. Our biophysical characterization narrows down the binding epitope between the interaction of dROS1 and BOSS. This could provide insight into the ligand binding region of mammalian ROS1. Indeed, for ALK, despite having structurally different ligands for different species, the binding regions are largely maintained[14]. However, alignment of the interreacting FNIII domain of dROS1 with mammalian ROS1 shows non conservation of the key hydrophobic residues in the peptide "FSI" (Supplementary Fig. 9b, c). This would indicate that if a ligand does bind at this site on ROS1, it likely uses a different set of interactions. Different ligands have been proposed to regulate mammalian ROS1 including NELL2[31] and Ribonuclease-7[32]. Both are soluble factors and not membrane anchored like BOSS. Therefore, it is possible that if ROS1 responds to soluble factors, it may use a different mechanism for binding and regulation.

Earlier efforts to elucidate dROS1's activation mechanism showed that neither a soluble monomeric ECR of BOSS nor a forced oligomer

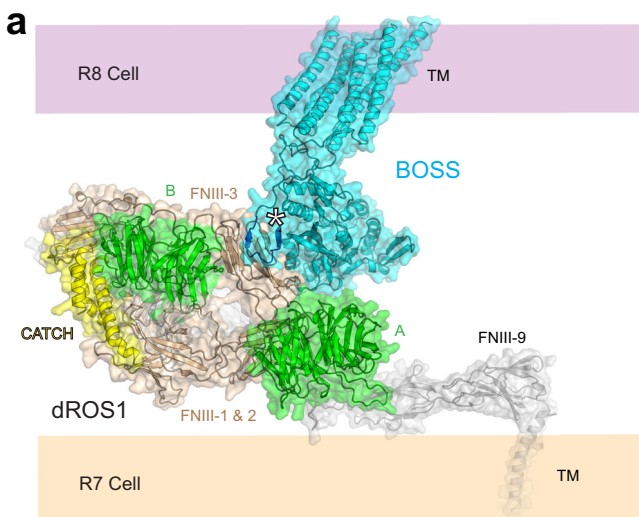

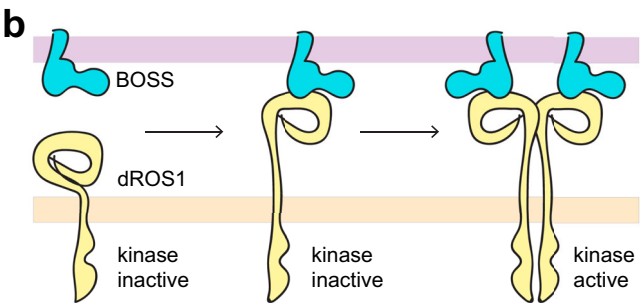

**Fig. 6 | Structural model of the dROS1 BOSS engagement complex. a** AlphaFold prediction model of dROS1 and BOSS binding at the surface of opposing cell membranes. The orientation of dROS1's ECR is similar to Figs. 2a and 4b. The model of the remaining ECR domains of dROS1 (gray) was generated from AlphaFold 3. The interaction site is indicated with a white asterisk, and blue strands for BOSS. **b** A schematic of the potential mechanism of dROS1 regulation by BOSS. BOSS binding shown stabilizing a conformation of dROS1 that can undergo receptor dimerization, leading to activation of the intracellular kinase domain. BOSS anchored to an opposing membrane may be necessary to transmit the force required for a dROS1 conformational change.

of BOSS' ECR is sufficient to stimulate dROS1's kinase activity[4,33]. Instead, a membrane anchored BOSS is required to induce R7 differentiation[33]. These findings, coupled with our AlphaFold predictions indicating BOSS is a monomeric protein, point to a mechanism whereby BOSS engagement of dROS1 from an opposing membrane may help coordinate a conformational change in dROS1 that is compatible with receptor dimerization and resulting activation (Fig. 6a, b). This would be similar to how EGF (Epidermal Growth Factor) regulates its receptor EGFR: a monomeric ligand induces receptor structural changes that relieve auto-inhibition and allow the receptor to dimerize[34]. Our structure, of dROS1's N-terminal ECR—being compact rather than linear, suggests that the bulky ECR could physically keep the kinase domain at distance and prevent oligomerization and signaling in the absence of ligand binding. An activation dependent conformational change in dROS1 may require tension afforded by both proteins being anchored in different membranes—and this tension is lost in soluble ECR constructs of BOSS whether monomeric or oligomeric, rendering them inactive[4,33].

A Dali[20,21] structural comparison dROS1's unique CATCH domain (including both helical hairpins) shows distant structural similarity to

the complement component protein C3a (Supplementary Fig. 10a, b). However, the disulfide bond connections are largely not maintained (Supplementary Fig. 10c, d). Notably, in the structure of the unprocessed complement C3 (PDB:2A73)[35], the C3a peptide interacts with the complex via the loop connecting H3-H4 (Supplementary Fig. 10e). This is equivalent to capping loop 2 of dROS1's CATCH domain. C3a also binds to and activates that C3a receptor (C3aR)—a GPCR[36,37]. Interestingly, the ligand for dROS1, BOSS, is an orphan GPCR. Therefore, given that dROS1's CATCH domain structurally mimics a GPCR ligand, this finding suggests that crosstalk between dROS1 and BOSS might be bidirectional. As the ECR of BOSS functions as a ligand for dROS1, so too the ECR of dROS1 might function as a ligand for BOSS. However, functional or evolutionary relatedness is not established, and further studies are needed to explore this possibility.

Future studies will focus on investigations with dROS1's full-length receptor to determine whether dROS1 resembles other well-known RTKs and cytokine receptors in their self-assembly upon ligand binding. While BOSS does not have a direct full-length homolog in humans, its TM regions show similarity to GPRC5B, and both respond to glucose stimulation[38]. However, GPRC5B completely lacks an ECR and could not function as a ROS1 ligand. More research should be done to determine whether dROS1 regulates this signaling function of BOSS. We acknowledge that during the review process, a related study by Cerutti et al. was published[39], and their Cryo-EM analysis identifies the same binding interface that we determined through HDX and modeling.

## Methods

### Recombinant protein expression and purification

BOSS ECR (residues 102-510) and dROS1 from *D. melanogaster* with various constructs dROS1 1-9 (residues 126-2113), 1–6 (residues 215-1795), 1–4 (residues 126-1298), 2-3 (residues 440-931), were incorporated into pFastBac plasmid (Invitrogen) with octa-histidine tag combined with a Factor Xa cut site at the N-terminus and Flag-tag at the C-terminus. The protein was then expressed in Trichoplusia ni (Hi5) cells driven by baculovirus infection at a density of $2 \times 10^6$ cells/mL. Mutants of BOSS and dROS1 were generated by site-directed mutagenesis using QuickChange Kit (Agilent Technologies). Proteins were purified from the medium (Ex-Cell405, Sigma-Aldrich) 2 days postinfection by directly flowing over Ni-Penta™ Agarose-Base Resin (Marvelgent Biosciences). Resins were washed 3 times with 35 mL of 20 mM imidazole (pH 8.0), 100 mM NaCl, and proteins were eluted first with 18 mL of 200 mM imidazole (pH 8.0), 100 mM NaCl and then with 18 mL of 500 mM imidazole (pH 8.0), 100 mM NaCl. Eluted proteins were further purified by size-exclusion chromatography (HiLoad 26/600 superdex 200 pg, GE Healthcare Life Sciences) in 20 mM HEPES (pH 7.5), 150 mM NaCl. Corresponding peaks were checked by SDS-PAGE (sodium dodecyl sulfate-polyacrylamide gel electrophoresis) gel and concentrated by Amicon Ultra-4 10 kDa MWCO concentrator (Millipore).

The His-tags of BOSS proteins and the mutant were cleaved by Factor Xa protease (New England Biolab) with 1 mM $CaCl_2$ and flowed over Ni-Penta™ Agarose-Base Resin (Marvelgent Biosciences). The flow-through was further purified by size-exclusion chromatography (Superdex 200 Increase 10/300 GL size exclusion column GE Healthcare Life Sciences) in 20 mM HEPES (pH 7.5), 150 mM NaCl. dROS1 (1-4) without his-tag was generated following the same protocol (for the HDX-MS experiment).

### CryoEM sample preparation and data collection

For cryo-electron microscopy, 3.5 μl of purified dROS1 (construct 1–6) at 500 nM were applied on glow-discharged Quantifoil Cu R1.2/1.3 300 mesh grids, and blotted for 5 s at 100% humidity and 16 °C. Samples were plunge-frozen into liquid ethane using Vitrobot Mark IV (FEI) and transferred into liquid nitrogen. Grids were imaged under the 300 kV

Titan Krios electron microscope (FEI) with a K3 summit direct electron detector (Gatan). Images were recorded under super-resolution mode at a pixel size of 0.825 Å/pixel using SerialEM v3.8.6 and Digital Micrograph v3.31.2359.03. Data were collected with a dose rate of 17.5 e-/pix/s for 0.06 s/frame. A total of 7835 images were recorded and processed using CryoSPARC v4.4.1[40,41]. Detailed parameters of data collection are summarized in Supplementary Table 1.

## CryoEM data processing and 3D reconstruction

Raw movies were motion-corrected using patch motion correction and contrast transfer function (CTF) parameters were estimated using patch CTF[40]. Bad micrographs were discarded by manual inspection. A small set of particles were manually picked, extracted and 2D classified as templates for autopicking. The template-picked particles were extracted with a box size of 786 Å and Fourier cropped to 192 Å, which was then used to generate 2D classes. Particles with good 2D classifications were re-cropped to 384 Å and re-extracted, which were then used for one round of ab initio reconstruction into 4 classes and iterative rounds of heterogeneous refinement into 5 classes. The best model (388,364 particles) was selected for non-uniform refinement. The overall resolution is 3.74 Å estimated by Fourier shell correlation (FSC) at 0.143 cutoff. These particle sets (388,364 particles) were then used in additional rounds of heterogeneous refinement into 4 classes, which gave us a 3.94 Å map (144,311 particles) with more density. These volumes were assessed by local resolution estimate using the CryoS-PARC v4.4.1[40] program and the final resolution was colored using UCSF ChimeraX. Both volumes were used in model building. Local mask was created covering the CATCH domain, FNIII-1, FNIII-2 and FNIII-4, from the 3.94 Å map using UCSF ChimeraX and imported back into CryoS-PARC v4.4.1[40]. Local refinement was performed using created local mask and yield a slightly more density around the masked area.

## CryoEM model building

The initial model of the structure was generated using AlphaFold2[16–18]. The model was segmented into individual domains and fitted into a map using ChimeraX. Given that the 3.74 Å map has a higher resolution, FNIII-1 to YWTD-B were fitted using this map. The cysteine-rich domain and FNIII-4 were fitted into the 3.94 Å map. Domains were then merged and imported into Coot 0.9.6[42] from SBGrid[43] along with the two maps for manual inspection. Real space refinement was done using Phenix 1.21-5207[44] from SBGrid[43]. Detailed statistics and parameters for model building and refinement are provided in Supplementary Table 1. Figures prepared with Adobe Illustrator 27.1.1.

## Surface plasmon resonance

Surface plasmon resonance (SPR) experiments of dROS1 BOSS interactions were all performed using BIAcore T200 in 20 mM HEPES (pH7.5), 100 mM NaCl. The Series S CM5 biosensor chip surface was activated with N- hydroxysuccinimide (NHS) and N-ethyl-N'-[3-(die-thylamino)propyl] carbodiimide (EDC). dROS1 constructs at 30 ng/μL in 10 mM acetate pH 4.0–6.0 were flowed over the activated surface at 10 μL/min for 300 s for covalent coupling. The remaining active sites were blocked by 1 M ethanolamine pH 8.5. Purified ligand BOSS proteins at varied concentrations were flowed over the dROS1 surfaces at 10 μL/min for 180 s. The response unit (RU) values at the plateau, corrected by subtracting background binding from the control surface, were used to measure of binding. Binding affinities were calculated by plotting the plateau RU values against BOSS concentrations and fitting the data to a single-site specific binding model using GraphPad Prism9. All binding curves were then plotted as BOSS concentrations versus maximal binding (Bmax).

## AlphaFold prediction

dROS1 model was searched from AlphaFold protein structure database (DeepMind and EMBL-EBI) and the construct 1–6 (residues 2156-1795)

was used to generate the model for CryoEM map. AlphaFold 3[19] (Alphafoldserver.com) was used for generating the prediction of full-length BOSS.

For the complex, initially, BOSS ECR (residues 102-510) and dROS1 construct 2-3 (residues 440-931) were used for heterocomplex prediction with AlphaFold 2 using the publicly available ColabFold v1.5.3[45]. This construct was chosen as it included the determined binding epitope and was of a size capable of running on the available resources. MSA options were set to 'mmseqs2_uniref_env' and pair mode was 'unpaired_paired'. All models converged on a binding epitope that only included the third FNIII domain of dROS1. However, the models exhibited a combination of parallel and anti-parallel binding modes. We then used AlphaFold 3 which allows for post translational modifications (glycosylations) to generate models of the complex between BOSS and dROS1's third FNIII domain. We added a single NAG (N-Acetyl-beta-D-glucosamine) to the predicted sites in BOSS (including those near the interface, N474 and N485). All resulting models converged with high confidence to the single complex used in this study.

For the full-length complex model we docked a full-length dROS1 model from AlphaFold 3 onto the complex, also AlphaFold 3, of full-length BOSS bound to the third FNIII domain of dROS1 (ipTM = 0.8, pTM = 0.68).

## Hydrogen-Deuterium Exchange and Mass spectrometry (HDX-MS)

Probing low-affinity interaction such as dROS1-BOSS poses a major challenge in HDX-MS experiments because dROS1-BOSS complex dilution with the deuterium labeling buffer (20 mM HEPES, 100 mM NaCl, pD 7.4) can lead to the complex dissociation. To maintain at least 80% of the complex after 10-fold dilution with the deuterium buffer, dROS1 at 0.6 mg/mL (4.5 μM) must be pre-incubated with a large excess amount of BOSS (210 μM, 11.8 mg/mL). Although such an experiment enables one to probe the complex, resulting mass spectra would be largely dominated by BOSS peptides due to the greater amount of BOSS, making dROS1 deuterium uptake assessment extremely challenging. Furthermore, protein sample overloading onto pepsin and UPLC columns can result in column overpressure and clogging. To circumvent these problems, we designed deuterium labeling experiment based on a previous study where the effect of weak EGFR kinase dimerization on the kinase activity was studied by immobilizing the his-tagged kinase domain on DOGS Ni-NTA lipid vesicles[46]. The immobilization of the his-tagged protein increases its local concentration such that the kinase domain with weak propensity to dimerize could be dimerized (the kinase remains monomeric up to 50 μM).

In our new deuterium labeling protocol, his-tagged dROS1 (construct 1-4) or BOSS (0.6 mg/mL, 100 μL) was immobilized on 50 μL packed Ni-Penta™ Agarose-Base Resin (Marvelgent Biosciences) for 10 min. Next, either untagged protein was co-incubated with the immobilized dROS1 or BOSS followed by gentle centrifugation at 30 x g for 30 s to remove the unbound protein in flow-through. Immobilized protein concentration on Ni-NTA and appropriate binding partner concentration were determined and optimized by eluting the immobilized protein in a quench buffer (200 mM glycine, 100 mM TCEP, pH 2.4) before proceeding to deuterium labeling experiments. For deuterium labeling, either immobilized dROS1 or BOSS (0.6 mg/mL, 100 μL) was co-incubated with untagged protein to form the complex, followed by a brief centrifugation (30 g for 30 s) to remove the unbound protein. Subsequently, the deuterium labeling buffer (50 μL, 20 mM HEPES, 100 mM NaCl, pD 7.4) was added to the Ni-resin and incubated for 10 min before the addition of cold 50 μL quench buffer (200 mM glycine, 100 mM TCEP, pH 2.4) to stop the labeling and elute the complex off the Ni-resin. The labeled sample was immediately flash-frozen in liquid $N_2$ and stored at −80°C until mass spectrometry analysis. Similarly, control deuterium labeling

experiments were carried out with either his-tagged dROS1 or BOSS only. A deuterium-labeled or -unlabeled protein was thawed quickly and immediately injected onto an Enzymate BEH pepsin column (Waters) maintained at 2 °C to digest proteins for 3 min in UPLC solvent A (water + 0.1% formic acid) at 100 µL/min. Peptic peptides were trapped in BEH C18 pre-column (2.1 × 5 mm, 1.7 µm, Waters), were separated on BEH C18 analytical column (1.0 × 100 mm, 1.7 µm, Waters), and eluted from the analytical column by 5% to 45% linear solvent B (acetonitrile + 0.1% formic acid) gradient over 7 min. HD-MS$^e$ data were acquired at 300–1200 m/z range using Synapt G2-Si mass spectrometer (Waters) using 0.4 s scan time in sensitivity ion mobility mode with Leu-Enk as a lock CCS and mass compound. Other instrument parameters were: capillary voltage, 3 kV; cone voltage, 10 V; source offset, 80 V; source temperature, 100 °C; desolvation temperature, 200 °C; cone gas flow, 0 L/Hr; desolvation gas flow, 700 L/Hr; nebulizer gas, 7.0 Bar. The low and high energy collision ramp was 5 to 10 V and 20 – 50 V, respectively.

### HDX-MS data analysis
ProteinLynx Global Server 3.03 (Waters) was used to sequence each peptic peptide, and the deuterium uptake of peptic peptides was evaluated using DynamX 3.0 (Waters), with the minimum intensity cut-off set to 1000, minimum product per amino acid to 0.1, and maximum MH$^+$ error set to 10 ppm. The deuterium uptake of both dROS1 and BOSS peptides is the average uptake of two biological samples ($n = 2$), with 3 technical repeats per biological sample. Peptides with a deuterium uptake standard deviation >0.5 Da from 2 biological repeat experiments were excluded from the analysis.

The HDX-MS data statistics, including the number of analyzed peptides, the peptide coverage, average peptide length/redundancy, and deuterium uptake standard deviations, as recommended by the HDX-MS guidelines[47], are presented in Supplemental Table 2 and the *Source data file*. To identify a binding region in dROS1 upon BOSS binding, the deuterium uptake of immobilized dROS1 in the presence of untagged BOSS was evaluated, and the uptake was compared to that of the his-tagged dROS1 without BOSS for obtaining the difference in the deuterium uptake (Δuptake). Similarly, to identify a binding region in BOSS upon dROS1 binding, the deuterium uptake of the immobilized BOSS in the presence of untagged dROS1 was assessed, and the uptake was compared to that of his-tagged BOSS without dROS1. The statistical significance for the sum of the Δuptake of each peptic peptide at 90% confidence limit was calculated as described previously[23]. All data were collected and analyzed according to consensus HDX-MS guidelines[47].

### Generation of Sevenless stable expressing cell line
The gene for the full extracellular domain of dROS1 (Sevenless) was fused with ALK's transmembrane and kinase domains into the pcDNA3.1 plasmid using the NEBbuilder HiFi DNA Assembly Cloning Kit (New England Biolabs). An NIH 3T3 cell line (ATCC CRL-1658) was used to generate a stable cell line of dROS1$_{ALK}$. This plasmid was transfected into NIH 3T3 cells with Lipofectamine 2000 (Thermo Fisher Scientific). The NIH 3T3 dROS1$_{ALK}$ stable cell line was selected with complete medium containing 400 µg ml$^{-1}$ G-418 (Santa Cruz Biotechnology).

### Cellular pull-down assay
Full-length BOSS, with a C-terminal hexahistidine tag was subcloned into a pVRC vector for expression. The Phusion Site-Directed Mutagenesis Kit (Thermo Fisher Scientific) was used to generate the BOSS DEQ mutant (designated as: BOSS$_{DEQ}$). dROS1$_{ALK}$ stable cells were seeded 2 days before transient transfection. The BOSS wild type plasmid or DEQ mutant plasmid was transfected into dROS1$_{ALK}$ stable cells using Lipofectamine 2000 (Thermo Fisher Scientific) and the cells were starved overnight after 24 h of transient expression. The

transfected cells were placed on ice and washed with ice-cold PBS twice. The cells were scraped and lysed with lysis buffer (20 mM HEPES pH 7.5, 150 mM NaCl, 1% NP-40) with 1× protease/phosphatase inhibitor cocktail (Cell Signaling Technology no. 5872) on ice. The lysed cells were then centrifuged at 15000 rpm at 4 °C for 15 min and supernatants were collected. 50 µL of Ni-Penta Agarose-Base Resin (Marvelgent Biosciences) was used for each pull-down sample. The resins were washed three times with 1 mL PBS and mixed with supernatant and incubated at 4 °C for 1 h with gentle rotation. The resin was then washed five times with 1 mL of lysis buffer for each round and bound proteins were eluted with lysis buffer containing 500 mM imidazole.

### Western blot
The pull-down samples were boiled in 1 × Laemmli sample buffer (Bio-Rad) and then separated on 4–12% Tris-Glycine gradient gel. The separated proteins were then transferred to PVDF membrane in transfer buffer (25 mM Tris, 200 mM glycine, 10% methanol (v/v)) for 49 min on ice. The membranes were first blocked with 5% milk in TBST buffer (20 mM Tris pH 7.5, 150 mM NaCl, 0.1% Tween 20) for 1 h at room temperature and then incubated with primary antibodies overnight at 4 °C. For detection of BOSS, a penta-his antibody (Qiagen, Cat. No. 34660) with 1:1000 dilution was used. The dROS1$_{ALK}$ receptor signal was detected by anti-ALK antibody (1:1000, Cell Signaling Technology, no. 3633). The receptor's phosphorylated signal was detected by anti-phospho ALK (Y1507) antibody (1:1000, Cell Signaling Technology, no. 14678). The membranes were then washed five times with TBST buffer and incubated with corresponding secondary antibodies. The HRP-linked anti-rabbit IgG (Cell Signaling Technology, no. 7074S) was used at a dilution of 1:4000. The HRP-linked anti-mouse IgG (Abcam, ab97046) was used at a dilution of 1:5000. The membranes were further developed with Amersham ECL Prime western blotting detection reagent (Cytiva Life Sciences) and imaged using a ChemiDoc (Bio-Rad).

### Reporting summary
Further information on research design is available in the Nature Portfolio Reporting Summary linked to this article.

## Data availability
The atomic coordinates for dROS1 have been deposited in the PDB with accession code 9BHX. The EM map has been deposited in the EMDB with accession codes EMD-44556. The monoisotopic mass (MH$^+$), centroid mass, deuterium uptake, and retention time of all peptides are detailed in the Source Data file. The HDX-MS data have been deposited to the ProteomeXchange Consortium via the PRIDE partner repository, with the dataset identifier; PXD051929. All other data needed to support the conclusions in this manuscript can be found in the main text or supplementary materials. All materials are available from the corresponding authors upon request. Source data are provided with this paper.

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

## Acknowledgements

We thank Claudio Alarcon, Yansheng Liu, Mark Lemmon, Kate Ferguson and their laboratories for valuable discussions. We additionally thank Yuhong Zuo and Long Han for their expertize in guiding the CryoEM processing. This work was supported by the NIH grants RM1GM149406 (D.E.K.) and NIH RO3-CA259881 (Y.T.).

## Author contributions

D.E.K. designed the overall project. J.Z., Y.T. and D.E.K. wrote the manuscript with input from all authors. J.Z. and D.E.K. analyzed the structure assisted by S.E.S. J.Z. generated all materials and performed all the experiments assisted by H.L., T.L., and Y.W. Y.T. designed, conducted, and analyzed the HDX experiment. H.L. carried out the cellular activation assays assisted by J.Z., S.L. and S.A.F.

## Competing interests

The authors declare no competing interests.
