## [Transparent Peer Review file · Nature Communications]

Structural Basis for the interaction between the *Drosophila* RTK Sevenless (dROS1) and the GPCR BOSS

Corresponding Author: Dr Daryl Klein

Version 0:

Reviewer comments:

Reviewer #1

(Remarks to the Author)

Zhang et al. investigated the structural basis for the interaction between dROS1 and BOSS by using CryoEM, HDX-MS, and computer simulation. This study is well-designed, but here are a few comments to improve the manuscript.

1. For the HDX-MS data, please provide the uptake plots (mean \pm S.D. from all the independent experiments) of all the regions that showed differences between different states. In addition, please provide HDX data table and HDX summary tables recommended by Masson et al. (Nature Methods, 2019, <https://doi.org/10.1038/s41592-019-0459-y>).
2. In page 3, (i.e. RXRR) should be (i.e., RXRR).
3. Please provide the full names of ALK and ALKAL.
4. Please compare the HDX heat map with the predicted structure of BOSS. This will increase the confidence for the predicted model structure. Likewise, please compare the HDX heat map with CryoEM structure of dROS1, which can provide more information about the conformational dynamics of dROS1.
5. The HDX was performed for only one time point (10 min), which is very rare. Please provide a rationale for performing only one time point.
6. The authors discussed that the binding of dROS1 to BOSS enhances structural disorder in a central beta-strand in the second lobe (residue 445-463) in the vicinity of the C-terminal beta-strand "VTNVTTY". However, the HDX difference at this region is near zero, not minus values (Suppl Fig 9b). The HDX differences near zero mean that the conformation has not been changed. Please correct if there are any errors in the manuscript.
7. The y-axis in Fig 4b or 4d is presented as stability index, and the y-axis in Suppl Fig 9b is Δ Uptake in Dalton. Is there any reason to present the y-axis in Fig 4b or 4d as stability index?
8. The authors described that HDX-MS was performed with 2 biological samples with 3 technical repeats. In Suppl Fig 9b, there are three dots for each peptide. What do these dots represent?
9. Fig 4b or 4d does not show any error bars. Please show error bars in these figures.
10. The authors reported, "dROS1SEQ construct was less stable and less pure than the WT construct". Thus, the inability of the dROS1SEQ construct to bind to BOSS might not be due to disruption of the binding interface but due to the instability of the construct. Therefore, further experiments are needed to confirm the binding interface.

Reviewer #2

(Remarks to the Author)

I co-reviewed this manuscript with one of the reviewers who provided the listed reports. This is part of the Nature Communications initiative to facilitate training in peer review and to provide appropriate recognition for Early Career

Researchers who co-review manuscripts.

Reviewer #3

(Remarks to the Author)

The manuscript by Zhang et al. presents a structural analysis of the extracellular domain of the *Drosophila* receptor tyrosine kinase dROS1, more commonly referred to as Sevenless, and its interaction with its ligand, Bride of Sevenless (BOSS). The key results of the study, and the methodologies with which they have been obtained are as follows:

1. Surface Plasmon Resonance is used to show that the full extracellular region (ECR) of dROS1 binds to the ECR of BOSS and to delimit a minimal region of the dROS1 ECR that also binds. The authors note that the K_d is relatively high, perhaps consistent with the fact that both proteins are membrane bound.
2. The authors express a construct including FN3 domains 1-6 and YWTD domains A,B, and C, and resolve a 3.94 angstrom structure using single particle CryoEM of most of the ECR represented in the construct. Within the structure, the authors note that the N-terminal region of the ECR adopts an interesting folded-over conformation stabilized which contains a novel cysteine-rich peptide that forms two disulfide stapled helical hairpins that associate with one another. Amino acid interactions that stabilize the folded-over structure are identified.
3. To identify likely points of interaction between dROS1 and BOSS, the authors utilize hydrogen-deuterium exchange and mass spectrometry (HDX-MS), a technology typically used to identify antibody/epitope interactions, and which the authors have previously used in studies of the RTK ALK. Applying this approach to the dROSS (1-4) construct (FN3 domains 1 to 4, YWTD domains A and B) and the full-length BOSS ECR, the authors describe several regions protected from exchange within BOSS, one of which (peptide 472-478), based on BOSS' structural prediction may represent a binding epitope for dROS1. Similarly, four regions protected from exchange are described within dROS1, one of which (peptide 878-896) is surface exposed and likely to be a site of interaction with BOSS.
4. The authors employ AlphaFold to generate a theoretical model of the complex between the ECRs of the two proteins, making use of their binding studies to constrain the predicted structures. A final model suggests a parallel beta-strand augmentation between dROS1's third FNIII domain and a C-terminal strand of BOSS' ECR, which is consistent with the location of the exchange-protected, presumed interacting epitopes that were identified by HDX-MS. To further validate the model for the complex, the authors introduce some relatively non-disruptive mutations into the FNIII domain of dROS1 and into the region of BOSS predicted to interact with FNIII at the corresponding region, finding that both of those mutations disrupt binding between the two ECRs, consistent with the model for the complex.

The conclusions outlined above are interesting and informative and the data supporting them are convincing and appear to be robust.

What about significance of the conclusions for this and related fields. Sevenless is an extremely significant protein in the history of *Drosophila* developmental studies. The role of Sevenless in the specification of the R7 photoreceptor cell in the fly eye is a classic paradigm of pattern formation in *Drosophila* development. Sevenless is also extremely important because genetic screens for dominant modifiers of hypomorphic or gain-of-function alleles of the sevenless gene were instrumental in identifying or confirming the roles of a number of other proteins as components of the Ras/Raf/Map Kinase cassette that operates downstream of many other RTKs. Finally, Sevenless is unique in that its activating ligand, BOSS, is a large transmembrane protein with features of a GPCR. This is made even more interesting because BOSS has been reported to respond to extracellular glucose and to regulate sugar and lipid metabolism in the fly. Thus, while the authors note that human ROS1 is the least well studied RTK, Sevenless is certainly not obscure. Accordingly, in view of Sevenless' historical significance and the unique nature of the receptor/ligand interaction, the work outlined above should be of relatively wide interest.

In fact, the authors do themselves a bit of a disservice in not including the name Sevenless in the title of their manuscript. I'd wager that many *Drosophila* investigators who would otherwise be extremely interested in this work might overlook the paper through not recognizing dROS1 as the name of the vertebrate orthologue of Sevenless (I would likely have been one of those). In my opinion, the authors also give rather short shrift to describing the historical/biological significance of Sevenless in the Introduction of their manuscript.

The manuscript is not without (a few) flaws. There are segments of the paper that are overly speculative, which detracts from the impact of the more solid components of the paper. Examples:

Regarding the author's reported similarities between the disulfide stapled helical hairpin structure of dROS1 and ALKSAL, and between the cysteine rich domain of dROS1 and complement protein C3a, insofar the authors' own data provides some fairly detailed information about the contacts that the two helical hairpins of dROS1 make with the 4th FNIII domain and the YWTD domain of dROS1 (and their conservation in other insect species), wouldn't speculation about meaningful similarities between this domain and those regions of ALKAL and C3a require some information about similar types of contacts that those helical hairpins make? As noted by the authors, the two helical hairpins in C3a do not even share the same disulfide bonding pattern as the cysteine rich domain of dROS1. Moreover, if I am reading this correctly in Supplementary Fig. 6a, the amino acid similarity between these regions in dROS1 and C3a is extremely limited. In the absence of clear functional or evolutionary relatedness between these domains in dROS1 and C3a, I question the rationale of naming this domain for C3a. dROS1 and C3a may have arrived at this similar-looking structure (albeit with useful features for the two proteins) by random chance.

A previous publication reported that BOSS' transmembrane segments suggest homology to metabotropic glutamate receptors (mGluRs) among Class-C GPCRs. The authors of this manuscript go on to obtain an AlphaFold prediction of the structure of the BOSS ECR which suggest that it adopts a single domain bi-lobed "Venus-Flytrap" architecture, also similar to mGluRs. They then performed a Dali structural comparison search, finding highest similarity to mGluR-2. The authors also note that BOSS lacks certain features normally found in mGluRs. They test whether the BOSS-dROS1 binding affinity is dependent upon the presence or absence of 2 mM L-glutamate and find no difference in binding. Whether BOSS binds to glutamate or acts as a glutamate receptor in *Drosophila* remains unclear. Based on comparison between the predicted BOSS ECR structure and the known mGluR-2, the authors also note the presence of a novel predicted N-terminal surface

loop in BOSS and speculate that this predicted loop might be critical for binding to dROS1. After performing mutagenesis to remove the loop, they observe no significant changes in binding between the mutated BOSS ECR and dROS1 and conclude that the unique N-terminal loop is not responsible for dROS1 binding. It is worth pointing out that this is all based on a structural prediction, that while amazingly accurate, AlphaFold does not accurately predict all protein structure, so the predicted novel loop may not even exist. However, these two sequences are examples of a recurrent theme that presents itself in this manuscript in which there is considerable speculation on a topic, some not very informative comparative analysis of structures (from a functional perspective), leading to an experiment to test the speculation, with a result indicating that the speculation was incorrect. Altogether, too much space in the manuscript is devoted to this and the authors should revise and streamline the manuscript to reduce the attention given to these less informative portions of the investigations/text.

There is another issue regarding BOSS' potential role as a GPCR. At the end of the discussion, the authors note the existence of a paper that indicates reports that BOSS functions as a glucose sensor (Kohyama-Koganeya et al., 2008). That manuscript reports that the transmembrane domain of BOSS shows similarity to vertebrate GPRC5B, recent studies of which seem to implicate the protein in lipid signaling and energy metabolism. The relationship between BOSS and GPRC5B appears elsewhere in the literature as well (see, for example, Hirabayashi and Kim, 2020, *The Journal of Biochemistry* 167:541-547, <https://doi.org/10.1093/jb/mvaa030>). I am not sure about the relationship between GPRC5B and the mGluRs but it would be worthwhile for the authors to discuss this discrepancy in a revised manuscript.

As noted above, this manuscript contains considerable interesting and informative data regarding the interaction between dROS1 and BOSS. However, there is still much to learn. It is important to note that the interactions between dROS1 and BOSS that are interrogated in this report are restricted to the two proteins' respective ECR regions. Insofar as most GPCRs are activated by ligands that interact with their TM domains, and given that activation of Sevenless requires the presence of the BOSS TM region, it is likely that interactions between Sevenless and the portion of BOSS not represented by its ECR are important. It is worth pointing this out more explicitly in the discussion.

A few additional minor points:

The text extending from line 60 to the end of line 63 is confusing. Isn't it accurate to say that the ECRs of most RTKs are in an inhibitory conformation for dimerization/activation in the absence of ligand? Or does this statement mean that while overexpression of other RTKs can overcome that inhibitory effect, overexpression of dROS1 or of the dROS1-EGFR-kinase chimera cannot? Please clarify.

Were constructs corresponding to FN3 domains 4-9, but lacking 2-3, tested for binding to BOSS? That is, are there other elements that bind to BOSS within the dROS1 ECR, distinct from the ones within 2-3?

Insofar as the CryoEM structure of dROS1's ECR is the most important piece of data in the manuscript, the schematic of the 1-6 construct, and its expression, should be shown in Fig. 1B.

Line 119. I think that you mean "perspective," not "prospective."

Lines 197/198. There is something (a clause?) missing from the statement that begins with "Whereas peptide....."

Line 289. "sufficient"

Reviewer #4

(Remarks to the Author)

This study by Zhang and colleagues focuses on structural and mechanistic understanding of activation of the receptor tyrosine kinase dROS1 (Sevenless), crucial for the differentiation of *Drosophila* R7 photoreceptor cells. dROS1 is activated through interaction with the extracellular region (ECR) of the GPCR BOSS (Bride Of Sevenless) on neighboring cells. While previous studies confirmed this activation and identified downstream signaling pathways, the structure of the dROS1-BOSS complex has been unknown. Using cryo-EM, the authors determine the first structure of the dROS1 ECR fragment, revealing a unique folded-over conformation. Based on HDX-MS analysis of the dROS1/BOSS ECR complex, they model the interacting regions between dROS1 and BOSS, and use these constraints to model the structure of the complex using AlphaFold. Mutagenesis was used to validate modeled interaction interface in the *in vitro* binding assays.

Overall, it is an interesting and impactful work. The main strength of this study is the novel structure for a *Drosophila* RTK that had not been previously characterized, nor had its homologs in other species, including humans. While the structure only corresponds to a fragment of dROS1 ECR, the authors show that this fragment is sufficient for BOSS binding and show that it has interesting structural features. These include the N-terminal Cysteine rich regions, which form a domain composed of two consecutive disulfide stapled helical hairpin structures. A similar domain was observed by the same authors in the previously published structure of an ALKAL - a ligand for the ALK RTK receptor. An interesting inference from this observation is that the authors hypothesize that dROS1 could be self-ligating, although this parallel is a bit hard to appreciate since they do not show how the ALKAL hairpin structure interacts with ALK receptor. Moreover, this N-terminal domain of dROS1 structurally also mimics a complement component protein C3a, which binds to C3aR GPCR. This finding prompts authors to hypothesize that in addition to BOSS being an activating ligand for dROS1, signaling might be bidirectional, and dROS1 might activate BOSS too, using its N-terminal domain. This is especially exciting because a ligand for BOSS has not been identified yet. The authors end up denoting the N-terminal domain of dROS1 as a CATCH domain, and thus, define a new domain fold.

Another significant impact of this work is advancing our understanding of the human ROS1 RTK. The dROS1 structure presented here provides valuable insights into this poorly characterized oncogene. While the nature of activating ligand for human ROS1 remains a big mystery, BOSS has been identified as the dROS1 ligand in flies. This study provides the first structural model for how BOSS and dROS1 interact, addressing a major gap in the field. An interesting, although highly speculative insight to a potential ligand for human ROS1 comes from the analysis of the AlphaFold prediction of BOSS structure reported here. The authors note BOSS's similarity to human metabotropic glutamate receptors (mGluRs), which hints to an exciting potential cross-talk between the mGluR and ROS pathways in humans (although not investigated in this paper).

The main weakness of the study is that it does not offer insights into how interactions between dROS1 and BOSS result in signaling. The binding interface is to some extent experimentally characterized but relies on creative solutions such as immobilization of the receptor domains on the surface of a resin for HDX-MS due to low binding affinity between dROS1 and BOSS, and utilization of AlphaFold predictions for BOSS structure and structure of the dROS1/BOSS complex. While based on these data, the authors succeed to identify regions critical for binding between the two receptors using their ECR fragments in vitro, an open question remains how this analysis translates to what determines the interaction between full-length receptors in cells. Most importantly, there are no data provided that test functional validation of the predicted interactions for signaling of dROS1. This is an important point that needs to be addressed to merit publication in Nature Communications.

Other comments:

1. Since the ECR structure here shown is compacted and as authors describe "self-ligated", could any of the HDX-MS data be interpreted in context of this structure opening to an extended state in the presence of BOSS?
2. The authors miss the opportunity to make parallels to the human ROS1 RTK. Can a homology model of its domain be made, and are residues in the relevant regions, involved in binding to BOSS, conserved? Some discussion of this would be useful.
3. It would be helpful to include the cartoons depicting domain composition of constructs used for structural analysis in Figure 2, and use them to explain the "folded-over" conformation.
4. Green letters marking relevant Cys residues in the alignment shown in Supplementary Figure 4 are not clearly visible.
5. Figure 3 needs better labeling of the new structural features (for example: N-terminal loop, C-terminal strand and loop are not explicitly labeled in this main figure, which would be helpful for readers). Figure 3, in general, could benefit from additional attention to coloring, improved visualization of depth, better labels, and overall refinement of the structural images. There is also a typo in Figure 3d; "stand" should be "strand".
6. Discussion of lack of conservation of the residues at the dimerization interface between BOSS and mGluRs is very abbreviated in the text, no specific residues are mentioned. The Figure 3e panel illustrating this point is quite underwhelming. The panel does not indicate which color corresponds to which receptor, and not all residues, including the coordinated glutamate, are labeled. It appears to be more like a first draft rather than a finalized figure.
7. Why 2mM glutamate was chosen for binding studies shown in Supplementary Fig.8? Could binding be still regulated, but at a higher glutamate concentration?
8. My suggestion is to make the orange region in Figure 4c, which depicts the HDX-MS mapped binding interface, darker or use a different color, as it is currently hard to see.
9. Some cartoon representations of the final receptor complex model(s) would be helpful and would tie the story back to cartoons introduced in Figure 1.

Reviewer #5

(Remarks to the Author)

I thoroughly enjoyed reading this paper, which presents the first experimentally (CryoEM) structure of the extracellular region of the dROS1 RTK that mediates ligand binding. The article is well written and superbly illustrated - two essential elements for good quality papers. The integrity and quality of the authors, who carefully documented all the results obtained, deserve admiration. I recommended this article for publication. Only careful re-reading for minor corrections is recommended.

Version 1:

Reviewer comments:

Reviewer #1

(Remarks to the Author)

Regarding HDX-MS data, there are a few serious issues that should be resolved.

1. Although the authors wrote "We have now included HDX-MS data, including the monoisotopic mass, centroid mass, deuterium uptake, and retention time without and with D2O and their respective standard deviations for all analyzed peptides, are provided as the source data in an Excel spreadsheet file", the reviewer cannot find these files in the submitted manuscript.
2. In the previous review, the reviewer asked to compare the HDX heat map with the predicted structure of BOSS and the

HDX heat map with the CryoEM structure of dROS1. However, the authors color-coded “the differences in deuterium uptake between the unliganded and liganded states (Figure S8)”. Figure S8 is informative, but this was not what the reviewer asked. If necessary, please perform back-exchange correction.

3. In response to the reviewer’s comment, “The HDX was performed for only one time point (10 min), which is very rare. Please provide a rationale for performing only one time point”, the authors responded “Our preliminary studies indicated that the 10-minute time point provided the most significant differences in deuterium uptake between the unliganded and liganded states of dROS1 and BOSS. For example, a previous study (Zhu S. et al. 2022, Biotechnol J 17(2) involving labeling range from 2 min to 240 min) and our previous study (Li, T. et al. 2021 Nature 600, 148-152, labeling range from 10 sec to 2 hours) show that deuterium uptake of protein-protein interaction surface remains relatively unchanged or shows <10 % exchange difference from seconds to 10-minute timescale” and “Due to sample limitations and the weak binding affinity between dROS1 and BOSS, we optimized our experiments to focus on this time point to maximize observable differences within practical constraints”. However, the previous study (Li, T. et al. 2021 Nature 600, 148-152) and many other HDX studies showed that the differences can be observed at earlier (10 s) or later (1000 s) time points. Moreover, for the weak binding proteins, earlier time points may be more relevant because 10-fold dilution with deuterium buffer can quickly dissociate the complex. Therefore, the reviewer strongly suggests performing HDX with more time points.

4. In the rebuttal letter, the authors stated, “This particular peptide had a very high standard deviation of >0.7 Daltons, so it was excluded from the analysis. We excluded peptides with higher than 0.5 Dalton SD from two biological repeat experiments, and this change does not change our initial finding that the additional β -strand in BOSS is likely to engage in dROS1 interaction, based on the protection from the exchange (Figure 4). We corrected main figure 4 and the text to reflect this change in the HDX-MS result section.”. It is not scientifically right to exclude data based on Standard deviation. The notion that the data showed high standard deviation may indicate that the experiments were not done correctly or the HDX mass spectra were not analyzed correctly.

Reviewer #2

(Remarks to the Author)

Reviewer #3

(Remarks to the Author)

I commend the authors for thoroughly addressing my concerns regarding the initial submission of this manuscript and can now recommend its publication.

Reviewer #4

(Remarks to the Author)

The authors have adequately resolved my concerns.

Version 2:

Reviewer comments:

Reviewer #1

(Remarks to the Author)

The authors have adequately resolved my concerns.

Reviewer #2

(Remarks to the Author)

Response to Reviewer #1

We thank the reviewer for their comments including “This study is well-designed, but here are a few comments to improve the manuscript.” We sincerely thank the reviewer for their expert suggestions and comments, which we believe has significantly improved our manuscript.

1. For the HDX-MS data, please provide the uptake plots (mean \pm S.D. from all the independent experiments) of all the regions that showed differences between different states. In addition, please provide HDX data table and HDX summary tables recommended by Masson et al. (Nature Methods, 2019, <https://doi.org/10.1038/s41592-019-0459-y>).

Thank you for this suggestion. We have now included HDX-MS data, including the monoisotopic mass, centroid mass, deuterium uptake, and retention time without and with D₂O and their respective standard deviations for all analyzed peptides, are provided as the source data in an Excel spreadsheet file. The data statistics table is also included as a Supplemental Table 2, as recommended by the HDX guideline Masson et al. (Nature Methods, 2019). We believe these additions enhance the transparency and reproducibility of our data.

2. In page 3, (i.e. RXRR) should be (i.e., RXRR).

Thank you for pointing out this typographical error. We have corrected it to "(i.e., RXRR)" in the revised manuscript.

3. Please provide the full names of ALK and ALKAL.

We apologize for the omission. We have now provided the full names upon their first mention in the manuscript: ALK (Anaplastic Lymphoma Kinase) and ALKAL (ALK and LTK Ligand). This clarification should help readers unfamiliar with these abbreviations.

4. Please compare the HDX heat map with the predicted structure of BOSS. This will increase the confidence for the predicted model structure. Likewise, please compare the HDX heat map with CryoEM structure of dROS1, which can provide more information about the conformational dynamics of dROS1.

Thank you for this valuable suggestion. We have now mapped the HDX-MS data onto the predicted structure of BOSS and the CryoEM structure of dROS1. The differences in deuterium uptake between the unliganded and liganded states are color-coded and overlaid on the AlphaFold-predicted structures in the revised Supplementary Figure S8. This visual representation highlights regions where solvent accessibility changes upon binding, enhancing the confidence in our predicted models.

Such analysis informs us about a change in structural flexibility/solvent accessibility of the same peptide region, however, the deuterium uptake of different peptide regions in the same protein cannot be compared unless the deuterium uptake of each peptide is corrected for the back exchange. This is due to the dependence of peptide deuterium uptake on the amino acid sequence/compositions, in addition to structural flexibility/solvent accessibility. Thus, more negative Δ raw uptake between the liganded and unliganded dROSS/BOSS in one region than that of another region does not indicate more structural stability in the former than the latter. For evaluating the solvent accessibility of the same region with and without a ligand, the raw deuterium uptake comparisons suffice as presented in Figure 4 and Supplemental Figure 8.

5. The HDX was performed for only one time point (10 min), which is very rare. Please provide a rationale for performing only one time point.

We appreciate your concern regarding the use of a single time point in our HDX-MS experiments. Our preliminary studies indicated that the 10-minute time point provided the most significant differences in deuterium uptake between the unliganded and liganded states of dROS1 and BOSS. Protein-protein interactions often involve large surface areas that lead to significant protection from exchange over longer timescales, and a single time point can be sufficient to detect these changes. For example, a previous study (Zhu S. *et al.* 2022, *Biotechnol J* 17(2) involving labeling range from 2 min to 240 min) and our previous study (Li, T. *et al.* 2021 *Nature* 600, 148-152, labeling range from 10 sec to 2 hours) show that deuterium uptake of protein-protein interaction surface remains relatively unchanged or shows <10 % exchange difference from seconds to 10-minute timescale. Due to sample limitations and the weak binding affinity between dROS1 and BOSS, we optimized our experiments to focus on this time point to maximize observable differences within practical constraints. Additionally, we have supplemented our HDX-MS data with complementary techniques, including structural modeling and mutagenesis studies, to validate our proposed binding site.

6. The authors discussed that the binding of dROS1 to BOSS enhances structural disorder in a central beta-strand in the second lobe (residue 445-463) in the vicinity of the C-terminal beta-strand "VTNVTTY". However, the HDX difference at this region is near zero, not minus values (Suppl Fig 9b). The HDX differences near zero mean that the conformation has not been changed. Please correct if there are any errors in the manuscript.

We thank the reviewer for finding the error. To clarify our HDX-MS results, we present the data in raw deuterium uptake of each peptide instead of the stability index shown in the previous version of our manuscript. All raw deuterium uptake data are listed in the source data file. The supplemental table lists the average standard deviation of all analyzed peptides based on the HDX guidelines (Masson, GR. *et al.* 2019 *Nature Methods* 16, 595-602).

7. The y-axis in Fig 4b or 4d is presented as stability index, and the y-axis in Suppl Fig 9b is Δ Uptake in Dalton. Is there any reason to present the y-axis in Fig 4b or 4d as stability index?

We thank the reviewer for pointing out our HDX-MS data ambiguities. We agree that using different y-axis units could cause confusion. We have revised Figures 4b and 4d to present the data as Δ Uptake in Daltons. The list of the deuterium uptake standard deviations of all analyzed peptides is included in the source data file. All HDX-MS data are now presented in the deuterium uptake, not in the stability index.

8. The authors described that HDX-MS was performed with 2 biological samples with 3 technical repeats. In Suppl Fig 9b, there are three dots for each peptide. What do these dots represent?

We appreciate the reviewer's inquiry. This particular peptide had a very high standard deviation of >0.7 Daltons, so it was excluded from the analysis. We excluded peptides with higher than 0.5 Dalton SD from two biological repeat experiments, and this change does not change our initial finding that the additional β -strand in BOSS is likely to engage in dROS1 interaction, based on the protection from the exchange (Figure 4). We corrected main figure 4 and the text to reflect this change in the HDX-MS result section.

9. Fig 4b or 4d does not show any error bars. Please show error bars in these figures.

The source data presents the deuterium uptake standard deviation for all analyzed peptides from two biological repeat experiments (not standard deviations of technical repeats). The new main figure 4 includes the calculated 90% confidence limit for the deuterium uptake difference using the average standard deviations of all analyzed peptides, as described previously (Houde, D., Berkowitz, SA., and Engen, JR. (2011) *J of Pharmaceutical Sciences* 100, 2071-2086). We also provide the supplemental table containing repeatability (deuterium uptake SD) in the dROS1:BOSS complex and respective unliganded protein.

10. The authors reported, “dROS1SEQ construct was less stable and less pure than the WT construct”. Thus, the inability of the dROS1SEQ construct to bind to BOSS might not be due to disruption of the binding interface but due to the instability of the construct. Therefore, further experiments are needed to confirm the binding interface.

We appreciate the reviewer’s comment regarding the potential instability of the dROS1SEQ construct affecting its binding to BOSS. Indeed, we were concerned about this possibility as well. To address this, we mutated the surface-exposed hydrophobic residues on BOSS that are predicted to interact with the FSI of dROS1’s third FNIII domain. This experiment was conducted as a reciprocal to the dROS1SEQ experiment to directly assess the binding interface from BOSS’ perspective. By performing mutations on BOSS, we aimed to validate the binding interface while minimizing the impact of the dROS1SEQ construct's instability, which could confound the results if only mutations on dROS1 were considered. This complementary approach was specifically included to strengthen the evidence for the binding interactions and to confirm the model, addressing the exact concern raised by the reviewer.

Response to Reviewer #2

We are grateful to reviewer 2 for their time and effort. Their co-review has substainally improved our manuscript.

Response to Reviewer #3

We sincerely thank the reviewer for their thoughtful and constructive comments, which have significantly improved our manuscript. We have addressed each point raised and detail our responses below.

1. Inclusion of "Sevenless" in the Title and Expanded Introduction

Reviewer comment:

"In fact, the authors do themselves a bit of a disservice in not including the name Sevenless in the title of their manuscript. I’d wager that many Drosophila investigators who would otherwise be extremely interested in this work might overlook the paper through not recognizing dROS1 as the name of the vertebrate orthologue of Sevenless (I would likely have been one of those). In my opinion, the authors also give rather short shrift to describing the historical/biological significance of Sevenless in the Introduction of their manuscript."

We appreciate the reviewer's suggestion and have revised the title to include "Sevenless":

Original Title: *Structural Basis for the Interaction between the RTK dROS1 and the GPCR BOSS*

Revised Title: Structural Basis for the Interaction between the RTK Sevenless (dROS1) and the GPCR BOSS

Additionally, we have expanded the Introduction to highlight the historical and biological significance of Sevenless in Drosophila developmental studies. Specifically, we added:

- A paragraph discussing Sevenless as a cornerstone in the study of Drosophila eye development and its role in understanding cell differentiation and signaling pathways.
 - Information on how genetic screens involving Sevenless led to the identification of key components of the Ras/MAPK signaling cascade, including Son of Sevenless (SOS).
-

2. Reduction of Speculative Content Regarding Structural Similarities

Reviewer comment:

"Regarding the authors' reported similarities between the disulfide stapled helical hairpin structure of dROS1 and ALKAL, and between the cysteine-rich domain of dROS1 and complement protein C3a... In the absence of clear functional or evolutionary relatedness between these domains in dROS1 and C3a, I question the rationale of naming this domain for C3a."

We agree with the reviewer that the naming based on structural similarities may be misleading without functional or evolutionary evidence. Therefore, we have:

- Revised the manuscript to reduce speculation about the functional similarities between dROS1, ALKAL, and C3a. C3a similarities and hypothesis was appropriately moved to the discussion section.
- Removed the designation of the domain based on C3a and instead introduced a new term, the **CATCH domain** (Cysteine-rich ALKAL-Type Coupled Helices), to reflect its structural features without implying unwarranted functional similarity.
- Clarified that the resemblance is primarily based on the rare structural fold of a disulfide-stapled helical hairpin, and how these folds interact with their receptors (Sup Fig. 5).

3. Streamlining Discussions on BOSS's Structural Predictions

Reviewer comment:

"...there is considerable speculation on a topic, some not very informative comparative analysis of structures (from a functional perspective), leading to an experiment to test the speculation, with a result indicating that the speculation was incorrect. Altogether, too much space in the manuscript is devoted to this and the authors should revise and streamline the manuscript to reduce the attention given to these less informative portions of the investigations/text."

We are grateful for this reviewer's comments. We have extensively revised the manuscript to make it more streamlined towards our conclusions, without diversions of incorrect hypothesis.

4. Addressing BOSS's Potential Role as a GPCR and Relationship to GPRC5B

Reviewer comment:

"There is another issue regarding BOSS's potential role as a GPCR... it would be worthwhile for the authors to discuss this discrepancy in a revised manuscript."

We have expanded the Discussion to address the relationship between BOSS and GPRC5B:

- Acknowledged previous reports suggesting homology between BOSS's transmembrane domain and vertebrate GPRC5B.
- Discussed recent studies implicating GPRC5B in lipid signaling and energy metabolism.
- Clarified the discrepancies between BOSS's potential homology to mGluRs and GPRC5B. This includes that BOSS' ECR has structural similarity mGluRs, whereas GPRC5B completely lacks an ECR.
- Highlighted that while BOSS shares some structural features with mGluRs, functional similarities in GPCR function may align more closely with GPRC5B, warranting further investigation.

5. Emphasizing the Importance of Transmembrane Interactions

Reviewer comment:

"...given that activation of Sevenless requires the presence of the BOSS TM region, it is likely that interactions between Sevenless and the portion of BOSS not represented by its ECR are important. It is worth pointing this out more explicitly in the discussion."

We have made explicit in the Discussion that:

- Our study focuses on the extracellular regions (ECRs) of dROS1 and BOSS.
- Interactions involving the transmembrane (TM) regions are likely critical for full receptor activation.

- Activation may involve additional conformational changes or interactions mediated by the TM domains, which were not captured in our current study.
 - Future work is needed to elucidate the role of the TM regions in dROS1 activation.
-

6. Clarifying Confusing Text and Addressing Minor Points

Reviewer comment:

"The text extending from line 60 to the end of line 63 is confusing... Please clarify."

We have revised the text to clarify that while overexpression of many RTKs can lead to ligand-independent activation due to increased receptor dimerization, overexpression of dROS1 or a chimeric dROS1 with the kinase domain of dEGFR does not result in activation without ligand, indicating a potent inhibitory role of dROS1's ECR.

Reviewer comment:

"Were constructs corresponding to FN3 domains 4-9, but lacking 2-3, tested for binding to BOSS? That is, are there other elements that bind to BOSS within the dROS1 ECR, distinct from the ones within 2-3?"

We attempted to express constructs corresponding to FN3 domains 4-9 but encountered issues with protein expression and stability. Our binding studies with the 2-3 construct, which showed similar affinity to the full ECR, suggest that the primary binding epitope resides within FN3 domains 2-3. This indicates that other regions are less likely to contribute significantly to BOSS binding.

Reviewer comment:

"Insofar as the CryoEM structure of dROS1's ECR is the most important piece of data in the manuscript, the schematic of the 1-6 construct, and its expression, should be shown in Fig. 1B."

We have updated Figure 2A to include the schematic representation of the 1-6 construct. And updated Figure 1B to show the expression data of the 1-6 construct, providing a clearer context for our binding and structural studies.

Reviewer comment:

"Line 119. I think that you mean 'perspective,' not 'prospective.'"

Thank you for catching this typographical error. We have corrected "prospective" to "perspective" in the manuscript.

Reviewer comment:

"Lines 197/198. There is something (a clause?) missing from the statement that begins with 'Whereas peptide...'"

We have revised the sentence for clarity:

- Original: "Whereas peptide 878-896 is more surface exposed and thus more likely to be a direct interaction site."
 - Revised: "Between these peptides, peptide 878-896 is more surface exposed and thus more likely to be a direct interaction site."
-

Reviewer comment:

"Line 289. 'sufficient'"

We have corrected the spelling of "sufficient" in the manuscript.

Response to Reviewer #4

We are grateful to the reviewer for their positive evaluation and constructive feedback. We have addressed the concerns and incorporated suggestions to strengthen our manuscript.

Major Concern: Functional Validation of Predicted Interactions

Reviewer comment:

"The main weakness of the study is that it does not offer insights into how interactions between dROS1 and BOSS result in signaling... Most importantly, there are no data provided that test functional validation of the predicted interactions for signaling of dROS1."

We acknowledge the importance of functionally validating our predicted interactions. To address this, we have:

- Conducted new cell-based experiments to assess the impact of binding site mutations on dROS1 activation.
 - Created a chimeric dROS1 receptor containing the human ALK kinase domain, allowing us to detect receptor activation using established antibodies.
 - Demonstrated that mutations disrupting the binding interface significantly reduce dROS1 binding and activation in response to BOSS in cells.
 - Included these results in the revised manuscript under the new section titled "**Impact of binding site mutations on dROS1 cell activation**" and updated Figure 5f accordingly.
-

Other Comments

1. Potential Conformational Changes in the ECR Upon BOSS Binding

Reviewer comment:

"Since the ECR structure here shown is compacted and as authors describe 'self-ligated', could any of the HDX-MS data be interpreted in context of this structure opening to an extended state in the presence of BOSS?"

We considered this possibility and re-analyzed our HDX-MS data. The data suggest that the interaction with BOSS does not induce significant global conformational changes in the ECR of dROS1. Instead, the protection patterns indicate localized binding without substantial unfolding or extension of the ECR. However, our HDX-MS experiments were designed and conducted to reveal epitopes of large protein-protein interactions and therefore may not be suitable for subtle dynamic changes over time. We have included this interpretation in the Discussion section.

2. Parallels to Human ROS1 RTK

Reviewer comment:

"The authors miss the opportunity to make parallels to the human ROS1 RTK. Can a homology model of its domain be made, and are residues in the relevant regions, involved in binding to BOSS, conserved?"

We have expanded the manuscript to discuss parallels with human ROS1:

- Performed sequence alignments between dROS1 and human ROS1, focusing on the binding interface.
 - Found that key residues involved in BOSS binding are not conserved in human ROS1.
 - Discussed the implications of these differences for ligand interactions and the potential for different regulatory mechanisms.
 - A supplementary figure (Supplementary Fig. 9b, 9c) shows the alignment and highlighting non-conserved residues.
-

3. Inclusion of Domain Cartoons in Figure 2

Reviewer comment:

"It would be helpful to include the cartoons depicting domain composition of constructs used for structural analysis in Figure 2, and use them to explain the 'folded-over' conformation."

We have updated Figure 2 to include domain composition cartoons for the constructs used. These visual aids help explain the folded-over conformation and provide a clearer understanding of the structural arrangement.

4. Visibility of Cysteine Residues in Supplementary Figure 4

Reviewer comment:

"Green letters marking relevant Cys residues in the alignment shown in Supplementary Figure 4 are not clearly visible."

We have adjusted the color scheme and font size in Supplementary Figure 4 to enhance the visibility of the green cysteine residues, ensuring they are easily distinguishable.

5. Improvements to Figure 3

Reviewer comment:

"Figure 3 needs better labeling of the new structural features... There is also a typo in Figure 3d; 'stand' should be 'strand'."

We have made significant changes and improvements to Figure 3 and extended analysis to Sup Fig. 7 as well. We believe this makes a clearer picture of the predicted structure of BOSS and how it compares to mGluRs.

- Added explicit labels for the N-terminus, C-terminal strand, and other structural features.
 - Enhanced coloring for better visualization.
 - Corrected the typo in Figure 3d from "stand" to "strand".
 - Improved the overall quality of the structural images.
-

6. Discussion on Conservation of Dimerization Interfaces

Reviewer comment:

"Discussion of lack of conservation of the residues at the dimerization interface between BOSS and mGluRs is very abbreviated... The Figure 3e panel illustrating this point is quite underwhelming."

We have expanded the discussion on this topic:

- Provided specific details about the residues involved in dimerization interfaces.
 - Clarified the lack of conservation between BOSS and mGluRs at these critical sites.
 - Updated Figure 3 panel of dimerization (now Supplementary Fig. 7) to:
 - Clearly indicate which colors correspond to BOSS and mGluR-2.
 - Label all relevant residues, and included the coordinated glutamate.
 - Enhance the visual quality to meet publication standards.
-

7. Rationale for 2 mM Glutamate in Binding Studies

Reviewer comment:

"Why 2 mM glutamate was chosen for binding studies shown in Supplementary Fig. 8? Could binding be still regulated, but at a higher glutamate concentration?"

We had selected 2 mM glutamate as an exceedingly high concentration based on standard physiological concentrations used in studies of glutamate receptors. Our experiments showed no significant difference in binding affinity in the presence of 2 mM glutamate. Our structural analysis makes it clear that BOSS is essentially pocketless and lacks all residues in mGluRs important for binding glutamate. While higher concentrations could be tested, the lack of effect at 2 mM suggests that glutamate does not influence the binding between dROS1 and BOSS. We have added this explanation to the Methods section.

8. Enhancing Visibility in Figure 4c

Reviewer comment:

"My suggestion is to make the orange region in Figure 4c, which depicts the HDX-MS mapped binding interface, darker or use a different color, as it is currently hard to see."

We have remade Figure 4 and adjusted the coloring and color contrast to represent the HDX-MS mapped binding interface, improving visibility and clarity.

9. Inclusion of Cartoon Representations of the Receptor Complex

Reviewer comment:

"Some cartoon representations of the final receptor complex model(s) would be helpful and would tie the story back to cartoons introduced in Figure 1."

We have added cartoon representations of the final receptor complex models in Figure 5 and the Discussion. These illustrations provide a visual summary of our findings and help connect the structural data to the overall activation mechanism proposed in Figure 6.

Response to Reviewer #5:

We are grateful to the reviewer for their comments, "I recommended this article for publication. Only careful re-reading for minor corrections is recommended."

We appreciate the reviewer's comments for improving the content of our manuscript. Our responses to the reviewer's concerns (written in blue text) are discussed below. We thank the reviewer for detailed discussions and suggestions.

Regarding HDX-MS data, there are a few serious issues that should be resolved.

1. Although the authors wrote “We have now included HDX-MS data, including the monoisotopic mass, centroid mass, deuterium uptake, and retention time without and with D₂O and their respective standard deviations for all analyzed peptides, are provided as the source data in an Excel spreadsheet file”, the reviewer cannot find these files in the submitted manuscript.

We apologize for our careless mistake in not uploading our source data file. The source data file should now be available together with the HDX statistics table (Supp Table 2). The listed data in the source data file show all recommended HDX-MS information based on the HDX guidelines (Masson GR. *et al.* 2019 Nature Methods, 16, 595-602.).

2. In the previous review, the reviewer asked to compare the HDX heat map with the predicted structure of BOSS and the HDX heat map with the CryoEM structure of dROS1. However, the authors color-coded “the differences in deuterium uptake between the unliganded and liganded states (Figure S8)”. Figure S8 is informative, but this was not what the reviewer asked. If necessary, please perform back-exchange correction.

We provided the HDX structure heat map (Figure S8).

The back exchange correction is necessary if one desires to compare the structural flexibility of one region with that of the other so that the structural flexibility of different regions can be compared on the same scale (%), independent of differences in the intrinsic exchange rate of peptides. As stated below, this was not the goal of our HDX experiment. Our goal is not to probe the dynamics of different regions and compare them. Instead, we asked a simple question, "which regions show reduced deuterium uptake when bound to a protein ligand?" Our HDX-MS is an epitope mapping experiment. Please see below for additional discussion regarding our HDX data collection strategy.

3. In response to the reviewer's comment, “The HDX was performed for only one time point (10 min), which is very rare. Please provide a rationale for performing only one time point.

Labeling time points are decided based on the goal of one's experiment. For instance, if one desires to compare the dynamics of different regions within a protein structure, obtaining multiple time points will be necessary for assessing the height of energetic barriers. In other words, the deuterium uptake at different time points evaluates the accessibility to exchange-competent structures, determined by the height of energetic barriers (at room temperature, the height is in the range of ~3-5 $k_B T$). This was not the goal of our HDX experiments. The goal of our experiment was to identify regions that show reduced deuterium uptake in the same regions (the same peptides) between the unliganded and liganded protein, without comparing the structural flexibility of one region (say region A) to that of the other region (region B) within the same protein state. HDX-MS epitope mapping does not require one to collect multiple labeling timepoints.

It is not rare to see comparisons of deuterium uptake at one time point. In fact, various deuterium uptake data figures in numerous literatures show figures with one-time point data (structure heatmaps, Woods plots, for example) although the uptake data at multiple time points are collected. In other words, it is rare to see HDX figures looking into deuterium uptake differences over time (say 10 sec and 10 min of the same peptide in the same protein state) and offer some structural interpretations based on the extent of the uptake increase over time. For example, when one shows a % exchange plot of a peptide with multiple labeling time points, what does the increase in deuterium uptake of a peptide from one time point to the other points mean? This aspect is rarely interpreted in literature in depth other than saying, "This region is flexible (or stable)."

We expect to see a deuterium uptake increase in our proposed binding regions over time (if multiple data points are taken) since thermal fluctuations will eventually allow all regions to undergo exchange. A key to a meaningful HDX-MS experiment is the selection of labeling time. For example, if multiple data points are taken at hours and days (rather than seconds to minutes), such data will not provide useful structural information.

If we were to take shorter labeling time data (10 sec, 1 min) and saw no change in the deuterium uptake between the liganded and unliganded states but observed the uptake changes at 10 min labeling time point (like our manuscript shows),

our current conclusion wouldn't change. Similarly, if the deuterium uptake data at 10 sec and 1 min between the liganded and unliganded states show reduced uptake, it only substantiates our findings.

As the reviewer pointed out in our 2021 Nature, the dimerization interface (PXL region) in GRD with CDX 125 shows protection only at hour-labeling timepoints. But, the PXL region is not a proposed binding site for CDX125. Our 2021 Nature paper describes CDX125 binding to the C-terminal loop region, occluding ALK binding (Figure 4b).

However, the previous study (Li, T. et al. 2021 Nature 600, 148-152) and many other HDX studies showed that the differences can be observed at earlier (10 s) or later (1000 s) time points.

As the reviewer stated here, changes in the deuterium uptake at different time points are expected since the deuterium uptake is cumulative over labeling time points unless significant back exchange occurs. The 10 s HDX labeling time point (the lowest energetic barrier height than the barrier height at the longer labeling time points) mainly probes loop regions since loops are inherently disordered. As seen in our previous 2021 Nature Figure 4a, any time points longer than 10 sec show protection from the exchange (GRD + CDX123) compared to the unliganded protein (GRD only). GRD with CDX125 shows protection at 1 and 2-hour labeling time points, but not as significant as CDX123 at 1 min, 10 min, and 1,2 hour. But, as described above, the PDX region is not a CDX125 binding site.

Since the goal of our HDX is not to compare the dynamics of different regions, we feel that one labeling time point data should suffice to inform us about possible binding interfaces. Our HDX results suggested two possible binding interfaces, including the helix at lobe 1 and 2 interface and β -strand 7, based on the reduced deuterium uptake upon ligand binding. To further narrow down the binding region, we employed AlphaFold 3 (Fig.5a and 5b) to obtain a binding model and validated it by the mutagenesis study (Fig.5e).

Moreover, for the weak binding proteins, earlier time points may be more relevant because 10-fold dilution with deuterium buffer can quickly dissociate the complex. Therefore, the reviewer strongly suggests performing HDX with more time points.

As the reviewer points out, the complex dissociation after the deuterium labeling buffer addition was a concern. This motivated us to devise the HDX protocol, as described in our method section. Ligand immobilization techniques, including SPR and BLI are widely used for probing protein-protein interactions. Those techniques can probe weak affinity interactions with sub- μ M K_D range partly because an immobilized ligand has high ligand density on an SPR chip or BLI probe. If the complex dissociated, we would not see changes in deuterium uptakes between the liganded and unliganded structures. Again, we are not addressing the dynamic question. HDX-MS data were done for the epitope mapping.

Performing more HDX data labeling time points does not circumvent the complex dissociation problem if the standard labeling protocol is employed (in which a pre-formed ligand-protein complex is labeled without immobilization). Regarding collecting more data points, we addressed the concern under the concern #3.

4. In the rebuttal letter, the authors stated, “This particular peptide had a very high standard deviation of >0.7 Daltons, so it was excluded from the analysis. We excluded peptides with higher than 0.5 Dalton SD from two biological repeat experiments, and this change does not change our initial finding that the additional β -strand in BOSS is likely to engage in dROS1 interaction, based on the protection from the exchange (Figure 4). We corrected main figure 4 and the text to reflect this change in the HDX-MS result section.”. It is not scientifically right to exclude data based on Standard deviation. The notion that the data showed high standard deviation may indicate that the experiments were not done correctly or the HDX mass spectra were not analyzed correctly.

We think, based on the author's experience, that the standard deviations (based on biological, not technical repeats) should be used as a guide to detect poor-quality spectra. We manually evaluate the quality of each peptide spectrum on DynamX, and ones with high standard deviations are almost always derived from poor-quality peptide spectra due to low peptide intensity counts. Low peptide intensity counts (noisy spectra) are mostly due to peptide hydrophobicity (leading to poor ionizability), poor pepsin digestion, and poor LC peptide separations. Also, long peptides (typically <20 residue long peptides) contribute to high standard deviations because the mass of such peptides (without deuterium, 0 sec time point) approaches the average molecular weight (rather than their monoisotopic mass), leading to peptide misassignment. The inclusion of peptide data with high standard deviations leads to erroneous interpretations. Thus, reporting high-quality spectra data ensures the accuracy of our interpretation and data reproducibility. Of note, a typical standard deviation range

(based on the average of biological repeats, not the average of technical repeats) is about ± 0.14 Da for small soluble proteins (Houde D, Berkowitz, SA., and Engen JR. 2011 J Pharma Sci 100, 2071-2086), and our reported HDX data sets show the standard deviations a little above this range (please see Supplemental Table).

Even if we included a peptide we removed (>0.7 Daltons, showing increased flexibility with a ligand), our conclusion would not change since we sought regions with increased protection upon a ligand binding. In summary, peptides with unusually high standard deviations (>0.7 Dalton, our cut-off is >0.5 Dalton) must be excluded from the analysis so that our interpretations are not skewed because of poor-quality peptide spectra. Since currently available HDX-MS processing software is extremely prone to peptide misassignment, manual inspection of each peptide spectrum is essential. In this process, standard deviations are extremely useful for spotting poor-quality spectra and misassigned peptides. It is ideal if one can make correct assignment of such spectra; however, since the number of manually inspected peptide spectra can be easily in the order of several hundred to a few thousand (depending on proteins), removing ones with high standard deviations and focusing on overlapping peptides with high-quality spectra are always the best strategy to save time and increase data confidence.